# Same Cause; Different Effects in the Brain

**Mariya Toneva**[*]                                                                                 MTONEVA@MPI-SWS.ORG
*Princeton University, Max Planck Institute for Software Systems*

**Jennifer Williams**[*] and **Anand Bollu**
*Carnegie Mellon University*

**Christoph Dann**
*Google Research*

**Leila Wehbe**
*Carnegie Mellon University*

**Editors:** Bernhard Schölkopf, Caroline Uhler and Kun Zhang

## Abstract

To study information processing in the brain, neuroscientists manipulate experimental stimuli while recording participant brain activity. They can then use encoding models to find out which brain "zone" (e.g. which region of interest, volume pixel or electrophysiology sensor) is predicted from the stimulus properties. Given the assumptions underlying this setup, when stimulus properties are predictive of the activity in a zone, these properties are understood to cause activity in that zone.

In recent years, researchers have used neural networks to construct representations that capture the diverse properties of complex stimuli, such as natural language or natural images. Encoding models built using these high-dimensional representations are often able to significantly predict the activity in large swathes of cortex, suggesting that the activity in all these brain zones is caused by stimulus properties captured in the representation. It is then natural to ask: "Is the activity in these different brain zones caused by the stimulus properties in the same way?" In neuroscientific terms, this corresponds to asking if these different zones process the stimulus properties in the same way.

Here, we propose a new framework that enables researchers to ask if the properties of a stimulus affect two brain zones in the same way. We use simulated data and two real fMRI datasets with complex naturalistic stimuli to show that our framework enables us to make such inferences. Our inferences are strikingly consistent between the two datasets, indicating that the proposed framework is a promising new tool for neuroscientists to understand how information is processed in the brain.

**Keywords:** encoding models, interpretability, neuroscience, fMRI, regression, neurolinguistics

## 1. Introduction

A major goal of neuroscience research is to understand how the brain processes information. To work towards this goal, neuroscientists often map where information is processed in the brain. This type of mapping is frequently learned from encoding models which predict brain measurements from the properties of a stimulus (Mitchell et al., 2008; Kay et al., 2008; Nishimoto et al., 2011; Huth et al., 2012; Wehbe et al., 2014a; Huth et al., 2016; Schrimpf et al., 2020). Since encoding models can be built to predict measurements at different scales (e.g. functional Magnetic Resonance Imagining

---

[*] Equal contribution.

. Code available at https://github.com/brainML/stim-effect

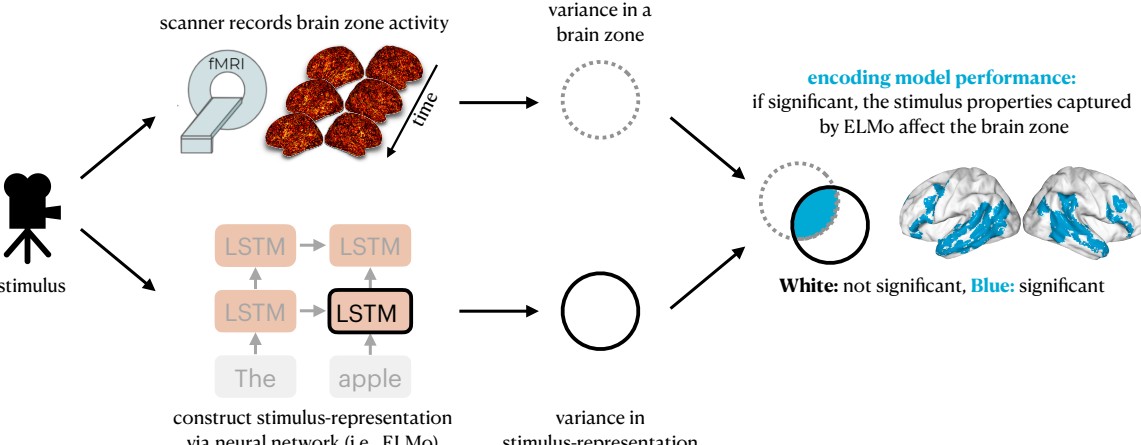

Figure 1: Using an encoding model to infer if stimulus properties captured by a stimulus-representation affect a brain zone. Brain activity is recorded during a passive experimental paradigm (i.e., movie viewing), and a stimulus-representation is constructed using a neural network. An encoding model that predicts the brain activity from the stimulus-representation is estimated (omitted from figure) and evaluated on held-out data and tested for significance.

(fMRI) volume-pixel (voxel) or region of interest (ROI), electrophysiology sensor, etc.) we use the term brain "zone" to agnostically refer to measurement locations at any of these scales.

In a passive experimental paradigm (such as movie viewing), the stimuli are chosen by the experimenter and precede the brain measurements. An encoding model can be used to identify the brain zones in which the activity is predicted by stimulus properties. An encoding model predicts brain activity as a function of a representation of the stimulus that corresponds to a specific hypothesis (e.g. a syntactic feature space can represent the linguistic structure of sentences and be used to identify brain zones processing syntax). Here, we use the term *stimulus properties* to refer to the real, latent features of the stimulus that affect brain activity, and the term *stimulus-representation* to refer to the vector representation that the experimenter builds to approximate the stimulus properties. It is assumed that the activity predicted using the encoding model is not related to some external factors or artifacts that are correlated with the stimulus. Under these assumptions, encoding models can be causally interpreted as revealing which brain zones are affected by stimulus properties (Weichwald et al., 2015) (see Fig.1). Note that, in this paradigm, we intervene on the stimulus and control the information presented to the participants. However, as we do not intervene directly on the brain (e.g., via transcranial magnetic stimulation) we cannot ask other, sometimes more crucial questions, such as whether the activity in a brain zone is *essential* for processing a stimulus.

If a stimulus-based encoding model tells us that two zones' measurements are affected by the stimulus properties, we can further increase our understanding of information processing in these zones by asking: do the stimulus properties affect both zones in the same way? This can be interpreted as asking if different subsets of the properties of the stimulus (e.g., in a movie stimulus, the language information and visual information) cause different effects in the two zones and/or if the same aspect of the stimulus (e.g. visual information) causes different effects in the two zones. As an illustration, consider two types of cells in the retina: on-center and off-center cells. Consider two such cells that process input from the same visual area (i.e., receptive field). Flashing a small light in the center of this shared receptive field will affect the response of both cells. However, the on-center cell's firing

rate will increase, while the off-center cell will be inhibited (Kuffler, 1953). The same cause leads to two different effects, and distinguishing these effects is key to understanding the function of the cells.

In settings with more complex stimuli (including naturalistic stimuli such as watching movies, reading books, listening to stories) it becomes hard to infer if the stimulus affects different zones in the same way as there are multivariate and often high-dimensional aspects to the stimulus. For instance, during natural reading, one zone can be sensitive to grammatical complexity while another can be sensitive to the presence of abstract meaning. If the stimulus-representation is rich enough to contain both grammatical and meaning information, both zones can be significantly predicted. In fact this difficulty is why it remains an open question if ROIs in the language network (Fedorenko et al., 2010) are affected in the same way by language stimuli (Wehbe et al., 2014a; Reddy and Wehbe, 2020; Caucheteux et al., 2021a; Huth et al., 2016; Deniz et al., 2019). Given that it is becoming increasingly popular in neuroscience to use more complex naturalistic stimuli that are more faithful to the real world (Sonkusare et al., 2019; Nastase et al., 2020; Hamilton and Huth, 2020), and to use complex representations from neural networks to build encoding models (Yamins et al., 2014; Wehbe et al., 2014b; Jain and Huth, 2018; Toneva and Wehbe, 2019; Wang et al., 2019; Schrimpf et al., 2020; Caucheteux et al., 2021b; Goldstein et al., 2021; Cross et al., 2021), it is important to enable such inferences in settings with complex stimuli.

More formally, we are interested in inferring if the causal effects $P(Y_1|do(S))$ and $P(Y_2|do(S))$ of the stimulus $S$ on two brain zones $Y_1$ and $Y_2$ are different. Generally, modeling the effect of a single cause over many outcomes fits under the umbrella of outcome-wide design (VanderWeele, 2017). Under our assumptions (the stimulus is set by the experimenter and no other artifacts affect both the stimulus and the brain activity), there are no confounders acting on both the stimulus and the brain activity, and the interventional distribution $P(Y_i|do(S))$ is equal to the conditional distribution $P(Y_i|S)$. This holds irrespective of any dependence between $Y_1$ and $Y_2$, and between these zones and other zones not included in the analysis. To illustrate this point for the reader, we derive in Appendix A the distributions $P(Y_i|do(S))$ for different Bayesian networks denoting the possible dependencies between $Y_1$ and $Y_2$, or $Y_1$ and $Y_2$ and other zones.

To make the inference of whether a stimulus affects two brain zones in the same way, we present a new framework that includes two new metrics. The first metric, **zone generalization**, captures the degree to which two zones are affected similarly by stimulus properties captured by a *stimulus-representation*. The second metric, **zone residuals**, captures the magnitude of any *stimulus-related* effect that is not shared between the two zones (even if this stimulus-related effect is caused by stimulus properties not included in the stimulus-representation). In Section 2, we present this framework as a decision tree. In Section 3, we introduce an implementation of both types of metrics. In Section 4, we present simulations that show that (1) when used together zone generalization and zone residuals enable researchers to infer if a stimulus affects two brain zones in the same way; and (2) our two proposed metrics provide new insights that current techniques for processing information in the brain cannot. We make our simulation data and code available so that researchers can test whether other implementations of these two metrics enable this inference (with their preferred statistical models). Lastly, in Section 5 we showcase the use of our framework on two fMRI datasets with complex naturalistic stimuli, and show when we can infer if a stimulus affects two brain zones in the same way and when this inference cannot be made. Our inferences generalize across these two datasets which capture different populations of participants and were acquired by different labs, in different countries, with different movie stimuli and different scanning parameters.

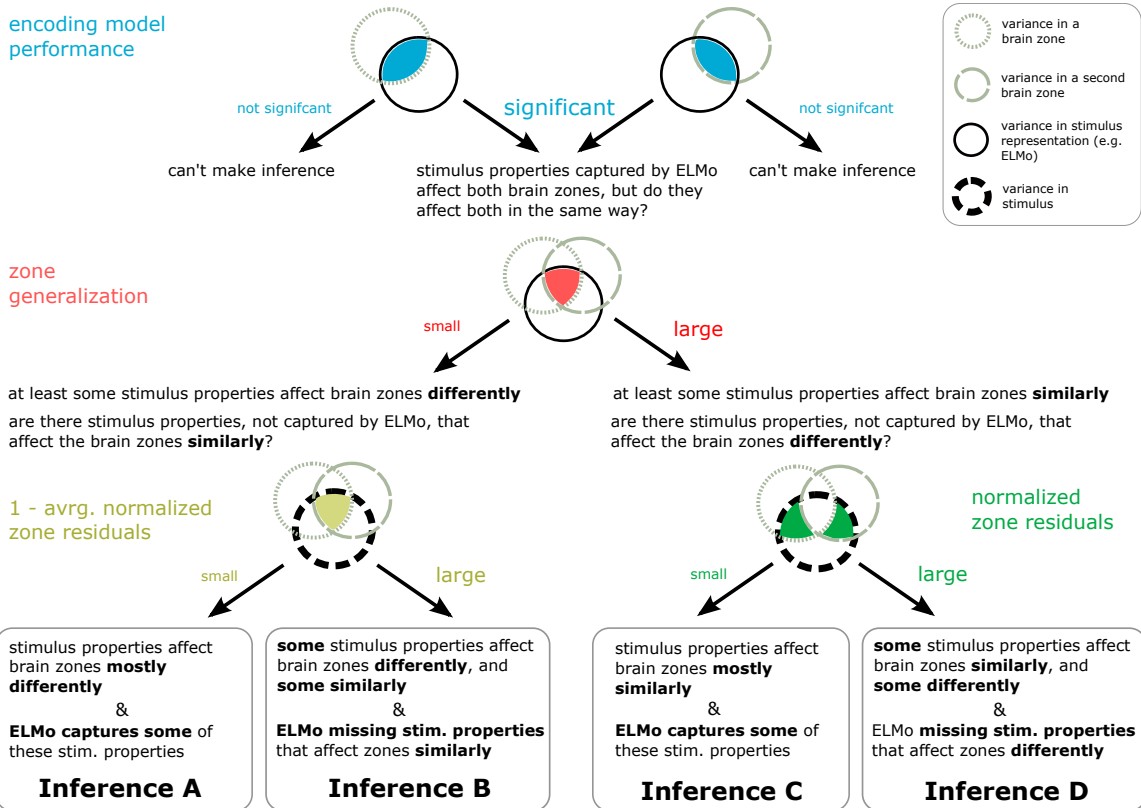

Figure 2: Proposed framework to infer whether two brain zones are affected by stimulus properties in the same way. The first step follows current work and computes the encoding model performance of predicting each of the brain zones as a function of a stimulus-representation (i.e. a representation from a deep neural network, such as ELMo (Peters et al., 2018) for language stimuli). We cannot infer directly from the encoding model performance whether the two brain zones are similarly affected by the stimulus, so we proceed to the second step. Large zone generalization allows us to infer that the brain zones are affected similarly by the stimulus properties captured by the stimulus-representation. At the last step, (normalized) zone residuals allow us to infer the magnitude of any stimulus effect that is not shared between the two zones, even outside of the stimulus-representation. We use both metrics to make one of four inferences.

## 2. Proposed Framework

Here we present the proposed framework as a decision tree (Fig. 2). A researcher first creates an encoding model to identify brain zones that are well predicted by a given stimulus-representation (e.g., ELMo (Peters et al., 2018)) (Fig. 1). Assume this approach reveals that the encoding model performance is significantly higher than chance. The typical encoding model pipeline stops here. In our framework, the researcher instead proceeds to computing zone generalization, measuring the degree to which two zones are affected in the same way by given aspects of a stimulus.

1. If zone generalization is large, we conclude that the brain zones are affected similarly by the stimulus properties captured in the stimulus-representation. It is still unclear however if the zones are affected similarly by the entire stimulus, or if the stimulus-representation is

incomplete and does not include stimulus properties that affect the zones differently. Here, we use our second metric, zone residuals, that captures the magnitude of any stimulus-related effect that is not shared between the two zones, even outside of the stimulus-representation:

   (a) If (normalized) zone residuals are small, we conclude that the two zones are affected mostly similarly by the stimulus, and that the stimulus properties that affect the zones similarly are (at least in part) captured by the stimulus-representation.

   (b) If (normalized) zone residuals are large, we conclude that the two zones are affected similarly by some properties and differently by others, however the stimulus-representation is incomplete and does not capture all the stimulus properties that affect zones differently.

2. If zone generalization is small, we conclude that the two zones are affected differently by the stimulus properties captured in the stimulus-representation. It is still unclear if the zones are affected differently by the entire stimulus, or if the stimulus-representation is incomplete and does not include all the stimulus properties that affect zones similarly. Here, we use our second metric again:

   (a) If (normalized) zone residuals are large, then we conclude that the two zones are affected mostly differently by the stimulus.

   (b) If (normalized) zone residuals are small, then we conclude that the two zones are affected similarly by some stimulus properties and differently by others, however the stimulus-representation is incomplete and does not capture all the stimulus properties that affect zones similarly.

In this work we employ linear functions in our metrics in order to relate them to the most frequently used type of encoding model—the linear encoding model. However, the framework is general as the metrics can be defined according to any predictive function.

## 3. Metric Definitions

A very commonly assumed encoding model for the effect of a stimulus on a brain zone is:

$$Y_i = g_i(X) + \epsilon_i, \tag{1}$$

where $Y_i \in \mathbb{R}$ is the observation at brain zone $i$ (e.g. fMRI voxel or ROI, EEG/MEG sensor, etc.), $X \in \mathbb{R}^d$ is the $d$-dimensional numerical representation of the corresponding stimulus (e.g. a word embedding obtained from inputting a word stimulus in a language model), $\epsilon_i \in \mathbb{R}$ to a noise term that is independent from the stimulus-representation and $g_i$ is a function describing the stimulus effect. However, the true underlying model of the stimulus effect is unknown. For some questions, such as whether a specific brain zone is affected by the stimulus (see Fig. 1), encoding models are sufficient and are widely used (Mitchell et al., 2008; Kay et al., 2008; Nishimoto et al., 2011; Huth et al., 2012; Wehbe et al., 2014a; Huth et al., 2016). For other questions, such as the one considered in the current work–whether a stimulus affects two brain zones in a similar way–we show that encoding model performance may not be sufficient on its own. Specifically, we show that when the true stimulus effect model differs from the one assumed by the encoding model, encoding model performance may lead to incorrect inferences. In contrast, the two metrics that we propose can help make the right inferences, even under such misspecifications.

Consider the following alternative stimulus effect model that allows for some of the effect to be caused by stimulus-related properties $Z$ that are not captured by the stimulus-representation $X$. This is a realistic assumption because stimulus-representations do not necessarily capture all brain-relevant information. Additionally, this model makes explicit that some of the stimulus effect is shared with another brain zone $j$ (i.e. the stimulus elicits an individual response and a shared response):

$$Y_i = \underbrace{g_i(X)}_{\text{individual signal of stimulus-representation}} + \underbrace{h_i(Z)}_{\text{individual signal of } Z} + \underbrace{\epsilon_i}_{\text{individual noise}} +$$
$$\underbrace{g_{ij}(X)}_{\text{shared signal of stimulus-representation}} + \underbrace{h_{ij}(Z)}_{\text{shared signal of } Z} + \underbrace{\epsilon_{ij}}_{\text{shared noise}} . \tag{2}$$

Next, we define the three metrics introduced in Fig. 2, and provide intuition about how the metrics would perform if the underlying model is indeed misspecified and is closer to the model in Eq. 2. In Section 4, we provide simulation results that support these intuitions quantitatively.

**Encoding model performance.** Following Eq. 1, an encoding model estimates for each zone the function $g(\cdot)$ associated with stimulus-representation $X$. Most commonly, this function is parameterized as a linear function (i.e. $g(X) = \langle X, \theta \rangle$, where $\theta \in \mathbb{R}^d$), and is estimated using a set of training observations for each zone. Encoding model performance in zone $i$ is often evaluated as the Pearson correlation of held-out data $Y_i$ and the corresponding predictions $\widehat{Y}_i = \hat{g}_i(X)$:

$$\texttt{encoding model performance}\,(\mathbf{zone}_i) = \text{corr}(\widehat{Y}_i, Y_i).$$

Intuitively, if the underlying stimulus effect model is the one in Eq. 2, the encoding model performance would reflect both the individual ($g_i(X)$) and shared ($g_{ij}(X)$) response due to the stimulus-representation. Thus, good encoding model performance could be due to either an individual response, a shared response, or some mixture of the two. Therefore, encoding model performance alone cannot reveal whether the stimulus properties captured in $X$ affect both brain zones similarly.

**First metric in framework: zone generalization.** Our goal is to infer whether a stimulus affects two zones in the same way. As we cannot use encoding model performance to make this inference, we propose *zone generalization*, which estimates the degree to which two zones are affected similarly by stimulus properties captured by the stimulus-representation:

$$\texttt{zone generalization}\,(\mathbf{zone}_i, \mathbf{zone}_j) = \text{corr}(\widehat{Y}_i, Y_j). \tag{3}$$

$\widehat{Y}_i = \hat{g}_i(X)$ is the prediction from an encoding model trained on zone $i$'s data (with $X$ corresponding to held-out stimuli). $Y_j$ is zone $j$'s data recorded when the same held-out stimuli was presented. Note that while zone generalization can be estimated in other ways, we chose this specific implementation because of its similarity to and ease of comparison with encoding model performance.

Intuitively, if the underlying stimulus effect model is the one in Eq. 2, the zone generalization would reflect the shared ($g_{ij}(X)$) response due to the stimulus-representation. Thus, large zone generalization indicates that at least some stimulus properties affect zones $i$ and $j$ in the same way. Small zone generalization, when accompanied by a significant encoding performance for both zones, indicates that at least some stimulus properties affect zones $i$ and $j$ differently. However, these possible inferences are fundamentally limited by how completely the stimulus-representation captures the stimulus properties. For example, if the stimulus-representation does not capture some stimulus properties that affect zones $i$ and $j$ similarly, zone generalization may be small.

**Second metric in framework: zone residuals.** Zone generalization enables us to infer whether at least some stimulus properties affect two zones similarly or differently, but it may be unable to present the complete picture if the stimulus-representation does not capture all stimulus properties. To address this limitation, we introduce *zone residuals*, which capture the stimulus-related variance that is unique to a brain zone. Our implementation of this metric builds on the intuition behind inter-subject correlation (Hasson et al., 2004) to estimate how much of the activity in zone $i$ (or $j$) is affected by the stimulus but not shared with zone $j$ (or $i$). Concretely, zone residuals are estimated between each pair of participants $S$ and $T$ and then averaged across all pairs:

$$\texttt{zone residuals}(\texttt{zone}_i, \texttt{zone}_j) = \frac{1}{M^2 - M} \sum_{S,T,S \neq T} \text{corr}(R_{i-j,S}, R_{i-j,T}), \qquad (4)$$

where $R_{i-j,P} = Y_{i,P} - Y_{j,P}\beta_P^{ij}$ is the residual of regressing $Y_{i,P}$ onto $Y_{j,P}$ and $\beta_P^{ij}$ is the ordinary least square solution. The residual $R_{i-j,P}$ corresponds to the activity in zone $i$ that cannot be predicted by the activity in zone $j$. We estimate how consistent this residual activity is across participants using Pearson correlation. Consistent brain activity between participants processing the same stimulus, assuming no artifacts are shared between participants, must be driven by that stimulus (Hasson et al., 2004; Hebart et al., 2018). Thus, zone-generalization estimates the amount of activity unique to zone $i$ that is stimulus-driven.

Intuitively, if the underlying stimulus effect model is the one in Eq. 2, zone residuals would reflect the individual response that is due to the whole stimulus ($g_i(X)$ and $h_i(Z)$). When paired with encoding model performance and zone generalization, zone residuals can help infer whether stimulus properties affect two brain zones similarly or differently, or whether the brain zones are affected similarly by some stimulus properties and differently by others.

### 3.1. Metric Normalization

Many of the metrics discussed in the previous subsections may be more interpretable when they are related to the amount of predictable brain activity (i.e. a "noise ceiling"). The noise ceiling we use here is the square root of inter-subject correlation, similarly to Wehbe et al. (2021). Inter-subject correlation reveals which zones are correlated across participants as they process the same stimulus, thus revealing zones that respond to the stimulus (Hasson et al., 2004). For each zone $i$:

$$\texttt{inter-subject correlation}(\texttt{zone}_i) = \frac{1}{M^2 - M} \sum_{S,T,S \neq T} \text{corr}(Y_{i,S}, Y_{i,T}). \qquad (5)$$

Whether the raw values for the individual metrics should be used or whether they should be put in perspective with a noise ceiling or a different quantity depends on the question under investigation. We refer the reader to Appendix B, where we include a figure and discussion to expand this point.

### 4. Simulations

Working with synthetic data where we know the ground truth for the stimulus effect model helps provide intuition for how informative each metric discussed in Section 3 can be in a variety of controlled scenarios. In this section, we analyze the performance of these metrics under the stimulus effect model presented in Eq. 2. To further highlight that our framework can be used under different

dependencies between the zones, we also analyze the performance of the metrics under two additional stimulus effect models, in which one of the brain zones acts as a cause of the other brain zone. We refer the reader to Appendix C.4 for additional details.

**Generating synthetic data.**    Consider an experiment where two participants are presented with the same naturalistic stimulus while their brain activity is being recorded at the same two brain zones. Suppose also that we decide on some numerical representation of the presented stimulus. To generate synthetic data, we make the simplifying assumption that all stimulus properties are captured by one of the following two disjoint representations: (i) $X \in \mathbb{R}^d$, the stimulus-representation itself or (ii) $Z \in \mathbb{R}^d$, a representation that captures all remaining stimulus properties that $X$ does not capture. We then synthesize brain activity measurements at each zone as a linear function of $X$, $Z$ and an additional term $\epsilon$ that represents stimulus-independent noise observed in the brain zone. Our data generation model includes mechanisms that allow us to vary how much of the stimulus-driven activity is due to both (1) the two brain zones responding to the stimulus properties similarly vs. differently and (2) stimulus properties captured by the stimulus-representations (vs. missing from). Details on these mechanisms and the overall data generation model are provided in Appendix C.

Figure 3:    Plotting average metric values under simulations that separate (Left) inference A or B from inference C or D and (Right) inference B or C from inference A or D.

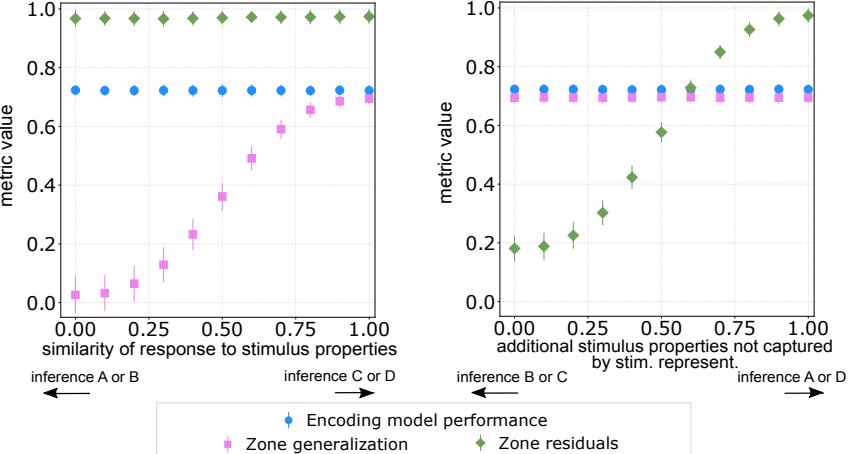

**Separating inferences A, B from inferences C, D.**    In the decision tree in Fig. 2, inferences A and B are separated from inferences C and D based on whether the two zones respond similarly to the stimulus properties captured in the stimulus-representation. We vary how similarly the two zones respond to the stimulus properties and observe the effect on our metrics. Fig. 3(Left) shows how each metric, computed and averaged over 1000 repetitions, varies as we do so. We find that encoding model performance and zone residuals remain relatively unchanged across the board. Only zone generalization seems to be informative of how similar two brain zones respond to stimulus properties, to help determine which pair of inferences (A or B vs. C or D) can be made.

**Separating inference A or D from inference B or C.**    From Fig. 2, one can separate inference A from B, and inference C from D, if one has information about the extent to which both zones respond to stimulus properties not captured by the stimulus-representation. In Fig. 3(Right) we plot metric values, computed and averaged across 1000 repetitions, as we vary the proportion of stimulus properties that are driving the brain zones but are not captured by the stimulus-representation. We find that of the three metrics of interest, only the zone residuals varies when this proportion is changed. This is because increasing the proportion of additional stimulus properties also increases

the proportion of stimulus properties that affect the zones differently, leading to larger zone residuals. Thus, after we use zone generalizations to narrow our search down to a pair of inferences (A or B vs. C or D), zone residuals can be used to precisely infer how the stimulus affects the pair of zones.

An important caveat to note is that zone generalization itself depends on the goodness of fit of the encoding model for the original brain zone. If the pair of zones we are investigating both have significant encoding model performances, zone generalization can be informative for inferring if the zones respond similarly to the stimulus properties captured by the stimulus-representation. But if the encoding model performance is non-significant for one or both zones, then we cannot even infer if the stimulus properties captured by the stimulus-representation affect both zones. Non-significant encoding model performance could be due to low SNR or an incomplete stimulus-representation that does not capture any of the stimulus properties that the zone(s) respond to.

Lastly, note that the halfway point between inferences A/B and inferences C/D occurs near $0.4$ zone generalization (Fig. 3(Left)), and the halfway point between inferences B/C and inferences A/D occurs near $0.6$ zone residual (Fig. 3(Right)). In the real fMRI data experiments in the next section, we use these values as the decision boundaries for the inferences depicted in Fig. 2.

## 5. Empirical Results on Two Naturalistic fMRI Datasets

Here, we examine if our proposed framework can help us infer if a stimulus affects different brain zones in the same way. We use two fMRI datasets with complex naturalistic stimuli (i.e., video clips), specifically because they present a trade-off between the number of participants and the amount of data recorded for each participant, enabling us to evaluate our framework under different constraints.

**HCP: short movies.** We use publicly available data from the Human Connectome Project (HCP) 7T dataset, with healthy participants between 22-36 years old (Van Essen et al., 2013). HCP fMRI data comes minimally pre-processed as FIX-Denoised data (Glasser et al., 2013; Griffanti et al., 2014; Salimi-Khorshidi et al., 2014). We analyze data from 90 participants that were randomly selected from the full dataset for exploratory purposes. Each participant watched naturalistic audio-visual video clips in English. In total, 60 minutes and 55 seconds of data were recorded per participant during 4 scans of approximately equal length. The fMRI sampling rate (TR) was 1 second.

**Courtois NeuroMod data: full-length movie.** The second fMRI dataset is provided by the Courtois NeuroMod group (Boyle et al., 2021). In this dataset, 6 healthy participants view the movie Hidden Figures in English. In total, approximately 120 minutes of data were recorded per participant during 12 scans of roughly equal length. The fMRI sampling rate (TR) was 1.49 seconds. We used the data release cneuromod-2020. This data is available by request at https://docs.cneuromod.ca/en/latest/ACCESS.html.

**Other data processing details.** For each participant we downsample the fMRI data by averaging the voxel activities within the 268 functionally defined ROIs from the Shen atlas (Shen et al., 2013; Finn et al., 2015), similarly to previous work (Rosenberg et al., 2015; Greene et al., 2018; Gao et al., 2019; Doss et al., 2020). For each participant this results in a dataset of dimensions number of TRs $\times$ 268 ROIs. These ROIs are entirely independent of our data as the Shen atlas was previously constructed from a separate group of healthy participants. Because we are interested in studying naturalistic language comprehension, we chose to identify language-relevant Shen atlas ROIs. We use ROIs localized for language comprehension by Fedorenko et al. (2010) and word semantics by Binder et al. (2009) (also defined entirely independently of our data). We consider a Shen atlas ROI to be a language ROI if $> 15\%$ of its voxels are within one of those previously identified ROIs. This procedure selects 55 Shen atlas ROIs (more details in Appendix D).

Figure 4: Encoding model performance at 34 significantly predicted ROIs (corrected at level 0.05).

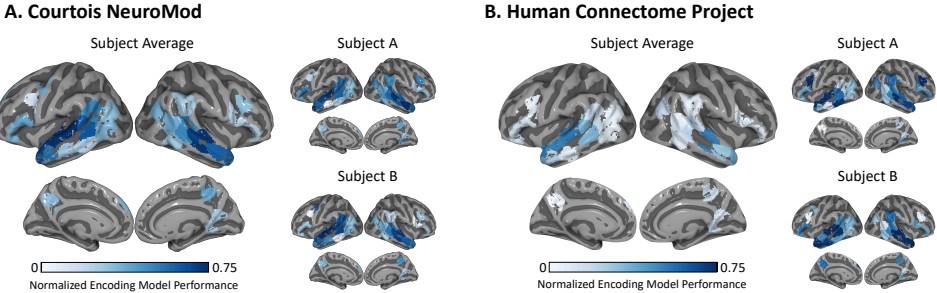

**Stimulus-representation.** The proposed framework is general and can be applied to a wide variety of stimulus-representations. Since we are interested in studying language processing, we obtain representations of the stimulus words by feeding the transcripts of the speech in the videos word-by-word into a pre-trained natural language processing model. We choose ELMo (Peters et al., 2018), a bidirectional multi-layer LSTM. Word representations obtained from the first hidden layer of the forward LSTM in ELMo, and contextualized with the previous 25 words, have been shown to significantly predict fMRI recordings of participants comprehending language (Toneva and Wehbe, 2019; Toneva et al., 2020). We focus our analyses on representations similarly collected from the first hidden layer of the forward LSTM of pre-trained ELMo (Gardner et al., 2017) when provided with chunks of 25 consecutive words.

**Encoding model performance.** As a first step in the framework, we identify the brain zones affected by the stimulus properties captured by ELMo. One encoding model is estimated independently for each ROI in each participant that predicts the recorded activity from the ELMo embedding. Models are estimated using ridge regression, following previous work (Nishimoto et al., 2011; Wehbe et al., 2014a; Huth et al., 2016; Toneva and Wehbe, 2019) and evaluated using cross-validation (CV), where one of the scans is heldout for testing during each CV fold (i.e. 4 folds for HCP, and 12 folds for Courtois NeuroMod). The regularization parameter is chosen by nested 10-fold CV.

We identify 34 bilateral language ROIs that are significantly predicted across participants in both fMRI datasets (one-sample t-test, FDR corrected for multiple comparisons across ROI at alpha level 0.05 (Benjamini and Hochberg, 1995)), suggesting that these ROIs are affected by the stimulus properties captured in ELMo. These results replicate previous findings that the language ROI are well predicted by representations from ELMo (Toneva and Wehbe, 2019; Toneva et al., 2020). For these 34 ROIs, we present the average normalized encoding performance across all participants in the datasets and for two representative participants in Fig. 4 (see Appendix Fig. 19 for additional participant-level performances). The encoding model performances are normalized by inter-subject correlation as described in Section 3.1.

**Zone generalization.** To infer if the stimulus properties captured by ELMo affect the 34 language ROIs in the same way, we turn to the next step of the framework and compute zone generalization. We present pairwise normalized zone generalization for the 34 language ROIs in both fMRI datasets in Fig. 5 (see Appendix Fig. 22 for the participant-level zone generalization). To visualize where the ROIs associated with each ROI pair are located on the brain see Appendix Fig. 17. The zone generalization values are also normalized by the inter-subject correlation. In both datasets, we find that there are large normalized zone generalization values (shown in red) which means at least some stimulus properties affect those ROI pairs similarly. These large normalized zone generalization values are consistent at the group and participant-level in both datasets. In both datasets, we also

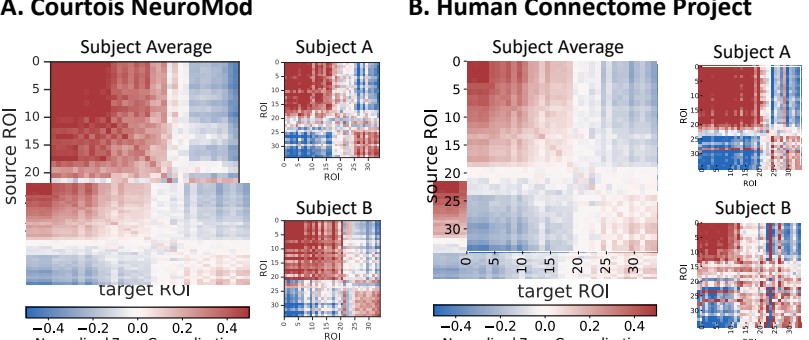

Figure 5: Zone Generalization. ROI pairs with large norm. zone generalization (red) are affected similarly by at least some stimulus properties. Pairs with large norm. zone generalization are consistent at the group and participant-level in both datasets.

find other pairs of ROI that exhibit small normalized zone generalization values, including negative values (see Appendix E for additional discussion about negative normalized zone generalization), which means that at least some stimulus properties affect those ROI pairs differently.

**Zone residuals.** At this point in the framework, it is unclear if ELMo is incomplete and does not capture some relevant stimulus properties. The last step is to compute the zone residuals. We present the normalized zone residuals for the $34$ language ROIs in both datasets in Appendix Fig. 18 (see Appendix Fig. 21 for participant-level normalized zone residuals). The zone residuals are normalized by the inter-subject correlation. In both datasets, we find large (dark green) and small normalized zone residuals at the group and participant-level. The ROI pairs with large normalized zone residuals are consistent at the group and participant-level in both datasets. Further, we empirically observed (see Appendix F), that the zone residual metrics was unstable when using less than 5 participants.

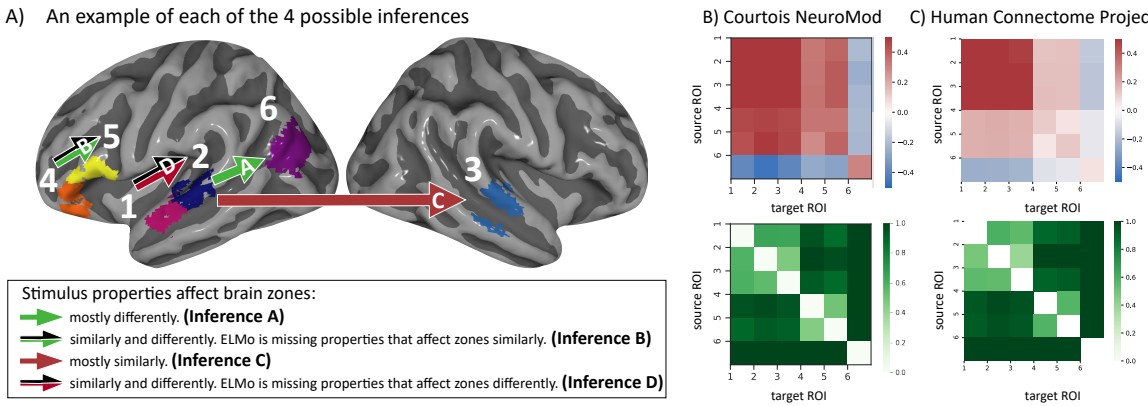

Figure 6: We use the proposed framework to infer answers to the question: Does the stimulus affect both brain zones the same way? We present an example of each of the four possible inferences.

**Examples of each inference type.** To make inferences A-D using the framework, we need to use encoding model performance, zone generalization and zone residuals. We focus on six language ROIs where it has been previously difficult to infer whether a naturalistic stimulus has the same effect (Reddy and Wehbe, 2020; Caucheteux et al., 2021a). We show an example of each of the four possible inferences that can be made between the six ROIs in Fig. 6. These relationships are consistent across both datasets (see Appendix Figs. 22, 23 for participant-level heatmaps). Since the relationships between ROI pairs can be asymmetric (ie. ROI $i$ could better generalize to ROI $j$ than ROI $j$ to ROI $i$), we depict relationships as a directed edge from a source to a target ROI. The

asymmetry in both zone generalization and zone residuals can be due to a difference in signal-to-noise ratio between the two brain zones, and also to an underlying mechanistic difference (e.g. a difference in the proportions of all stimulus properties that similarly affect the zones; see Appendix G).

Inference A, between ROI 2 (left posterior temporal gyrus) and ROI 6 (left angular gyrus) indicates that while they are both significantly predicted by ELMo, they are affected differently by language properties captured by ELMo. Inference B between ROI 4 (left inferior frontal gyrus, pars orbitalis) and ROI 5 (left inferior frontal gyrus) indicates that both ROIs are similarly affected by some language properties that are not included in ELMo. Inference D between ROI 1 and ROI 2 (anterior and posterior left temporal gyri) indicates that while these ROIs are affected similarly by some properties, ELMo is still missing some language properties that affect the zones differently. Finally, inference C between ROI 2 and ROI 3 (left and right posterior temporal gyri) supports the hypothesis that some language properties similarly affect the bilateral posterior temporal cortex.

**Discussion.** In contrast to encoding model performance alone, our proposed framework can be used to make inferences A-D as outlined in Fig. 2. Note that in inference B both zone generalization and zone residuals are small, and in inference D both zone generalization and zone residuals are large. In both cases, the stimulus-representation does not capture the relevant stimulus properties and our modeling can be improved by adding a stimulus-representation that does. For example, our results suggests that ROI 1 and ROI 2 (anterior and posterior left temporal gyri) in Fig. 6 may be easier to distinguish if we add stimulus-representations that capture the unique properties that either ROI is affected by. Our framework can be used as a test for future stimulus-representations—if these stimulus-representations lead to significantly larger encoding model performance for either ROI 1 or ROI 2, while the zone generalization and zone residuals do not change significantly, then the new stimulus-representation better captures some of the unique properties that affect the respective ROI.

## 6. Related Work

To investigate if brain zones respond similarly or differently to the stimulus, researchers have previously used encoding model weights (Çukur et al., 2013; Huth et al., 2012, 2016; Deniz et al., 2019). Encoding model weights can reveal the latent directions of largest variance in the tuning to a stimulus-representation (Huth et al., 2012, 2016; Deniz et al., 2019) and even how tuning in the same zone is affected by a specific task (i.e., attending to humans) (Çukur et al., 2013). The latent directions of largest variance can be revealed by obtaining the principal components of the brain zone weights and plotting the first few principal component scores on the brain (Huth et al., 2012, 2016; Deniz et al., 2019). The change of tuning in the same zone due to attention can be revealed by correlating a zone's encoding model weights with a binary template of the attended stimulus (Çukur et al., 2013). These approaches thus evaluate changes in tuning qualitatively (by producing interpretable brain maps) and quantitatively (by computing metrics on the weights).

Zone generalization estimates tuning differences not through the weights but through the prediction of new data. It could be considered a more stringent method because it ignores differences in weights that don't affect generalization performance. Zone generalization is a general metric that builds upon temporal generalization (King and Dehaene, 2014) and spatial generalization (Toneva et al., 2020), which can be considered two specific instances of this metric. King and Dehaene (2014) proposed temporal generalization to infer if the pattern of responses to a stimulus at a given point in time is similar to that at another point in time; subsequently this metric was used in multiple works (Blanco-Elorrieta and Pylkkänen, 2017; Hebart et al., 2018; Fyshe et al., 2019; Fyshe, 2020). Toneva

et al. (2020) adapted temporal generalization to compare the pattern of stimulus responses in different voxels. To the best of our knowledge, there are no existing metrics that are similar to zone residuals.

Other previous works that study information processing in the brain can be broadly classified in two groups that focus on relationships between: 1) two zones, or 2) a brain zone and a stimulus. Along the first direction, the most common approach is functional connectivity (Friston et al., 1993). Most frequently, functional connectivity correlates the measurements between two zones (Mohanty et al., 2020) of the same participant. Two brain zones can be correlated due to various factors beyond the stimulus, so functional connectivity does not capture whether two brain zones are responding similarly to the same stimulus (see simulation results in Appendix C.5). Psychophysiological interactions (PPI) (Friston et al., 1997) improves on functional connectivity by considering how the functional connectivity between brain zones differs under different experimental conditions. However, this metric still only quantifies whether two brain zones are related with respect to a stimulus, rather than whether they respond to the stimulus in the same way.

Along the second direction, the most common approaches are encoding models, inter-subject correlation (ISC), and representational similarity analysis (RSA) (Kriegeskorte et al., 2008). We discussed the relationships of our metrics to encoding models and ISC at length in Sections 1 and 3. RSA measures the similarity of two representational dissimilarity matrices, each describing the distances between pairs of representations of stimuli (e.g. a stimulus-representation or a brain zone). RSA between a brain zone and a stimulus-representation bears similarity to encoding model performance—they both estimate the strength of the relationship between the zone and the representation, though the RSA value is less readily interpretable. Like encoding model performance, standard RSA does not distinguish between the four inferences (see simulation results in Appendix C.5). A more complicated setup could be devised in which RSA is computed between predictions $\hat{Y}_1$ and real data $Y_2$, or between the zone 1 and zone 2 residuals of two participants. We consider such setups to be alternative implementations of our zone generalization and zone residuals metrics.

## 7. Conclusion and Future Work

We presented a new framework including two metrics, zone generalization and zone residuals, to enable researchers to infer if a stimulus affects brain zones in the same way. We showed in simulation that, when used in addition to significant encoding model performance, zone generalization and zone residuals enable this inference, while commonly used methods for studying information processing do not. Finally, we showed that the results from our framework generalize across two naturalistic fMRI datasets which capture different populations and are acquired by different labs in different countries with very different experimental setups and scanning parameters.

To make inferences using the fMRI datasets, we base the thresholds for the decision points illustrated in Fig. 2 on the simulated data. While these thresholds lead to repeatable results across two real datasets, future work that establishes principled dataset-specific thresholds may be fruitful. One idea is to make the decisions based on significance, similarly to the first step of the framework. Because there are not good priors for the chance values of zone generalization and zone residuals, the null distributions can be estimated using permutation tests. The implementation details of these tests (e.g. how they are performed and aggregated across participants) can be defined in future work. Overall, our proposed framework is a tool for computational neuroscientists who are interested in understanding how information is processed in the brain.

## Acknowledgments

The Courtois project on neural modelling was made possible by a generous donation from the Courtois foundation, administered by the Fondation Institut Gériatrie Montréal at CIUSSS du Centre-Sud-de-l'île-de-Montréal and University of Montreal. The Courtois NeuroMod team is based at "Centre de Recherche de l'Institut Universitaire de Gériatrie de Montréal", with several other institutions involved. See the cneuromod documentation for an up-to-date list of contributors (https://docs.cneuromod.ca). Additional data were provided in part by the Human Connectome Project, WU-Minn Consortium (Principal Investigators: David Van Essen and Kamil Ugurbil; 1U54MH091657) funded by the 16 NIH Institutes and Centers that support the NIH Blueprint for Neuroscience Research; and by the McDonnell Center for Systems Neuroscience at Washington University. Research reported in this publication was partially supported by the National Institute On Deafness And Other Communication Disorders of the National Institutes of Health under Award Number R01DC020088. The content is solely the responsibility of the authors and does not necessarily represent the official views of the National Institutes of Health. Research reported on this paper was also partially supported by a Google Faculty Research Award to L.W., Carnegie Mellon University's Center for Machine Learning and Health fellowship to J.W., and Ruth L. Kirschstein National Research Service Award Institutional Research Training Grant T32MH065214 and Princeton University's C.V. Starr Fellowship awarded to M.T. We thank Edward Kennedy and Aaditya Ramdas for useful discussion.

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

## Appendix for Same Cause; Different Effects in the Brain

### Appendix A. Estimating Causal Effect of the Stimulus on Each Brain Zone

As we have stated in the Section 1, under our assumptions, there is no confounder that affects both the stimulus and the brain activity in a zone. This means that $P(Y_1|do(S)) = P(Y_1|S)$ and $P(Y_2|do(S)) = P(Y_2|S)$ regardless of the causal relationships between $Y_1$ and $Y_2$ or the presence of additional zones that have an effect on one or both zones. As an illustration for the reader, we show here what happens under various configurations for the two zones and three zones case.

We consider possible DAG configurations for three brain zones and a stimulus, under our paradigm and assumptions specified in Section 1. As we only apply our zone residual and zone generalization metrics to pairs of brain zones that are affected by the stimulus, we only consider configurations where at least two brain zones are affected by the stimulus. If the third brain zone ($Y_3$) does not have a causal relationship with $Y_1$ nor $Y_2$ (i.e., $Y_3 \perp\!\!\!\perp Y_1|S$ and $Y_3 \perp\!\!\!\perp Y_2|S$) then the configuration simplifies to the two brain zone setting shown in Fig. 7. In the Fig. 7A configuration we can use the truncated factorization (Pearl and Verma, 1991; Pearl, 2000) (also known as the manipulation theorem (Spirtes et al., 1993), and implicit in the G-computation formula (Robins, 1986)) to show that $P(y_1|do(S = s)) = P(y_1|s)$. Another, perhaps more intuitive, way to think about this is that $P(y_1|do(S = s)) = P(y_1|s)$ if $Y_1$ is a direct effect of $S$ and the two variables do not have an observed or unobserved confounder. This similarly holds for $Y_2$.

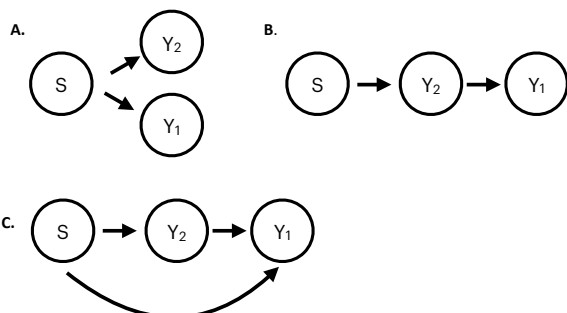

Figure 7: Causal DAG configurations for two brain zones and a stimulus. Under our paradigm and assumptions, these are all the possible configurations, where the stimulus $S$ affects both $Y_1$ and $Y_2$, up to permutations of the brain zones.

In the Fig. 7B configuration we show that $P(y_i|do(S = s)) = P(y_i|s)$. For $Y_2$ the truncated factorization shows that $P(y_2|do(S = s)) = P(y_2|s)$. For $Y_1$ we use the truncated factorization and then marginalize over $Y_2$ as follows:

$$P(y_1|do(S = s)) = \int P(y_1|y_2)p(y_2|s)\,dy_2$$
$$= P(y_1|s).$$

This holds similarly for the case where nodes $Y_1$ and $Y_2$ are permuted.

In the Fig. 7C configuration we also show that $P(y_i|do(S = s)) = P(y_i|s)$. For $Y_2$ we can again use the truncated factorization to show that $P(y_2|do(S = s)) = P(y_2|s)$. For $Y_1$ we can show that

this is the case using the truncated factorization and marginalization over $Y_2$:

$$P(y_1|do(S = s)) = \int P(y_1|y_2, s)p(y_2|s)\, dy_2$$
$$= P(y_1|s).$$

This holds similarly for the case where node $Y_1$ and $Y_2$ are permuted.

If the third brain zone ($Y_3$) only has a causal relationship with $Y_1$ or $Y_2$ then the configuration is as shown in Fig. 2A-B, up to the permutation of the zones. Here we also show that $P(y_i|do(S = s)) = P(y_i|s)$. For Fig. 8A-B $Y_3$ and $Y_2$ the truncated factorization directly shows this. For Fig. 8A $Y_1$ this is the case because this is a generalization of the calculation for $Y_1$ in Fig. 7B, where $Y_2$ has been replaced by $Y_3$. For Fig. 8B $Y_1$ we can show this using the truncated factorization, definition of conditional independence and marginalization over $Y_2$ and $Y_3$:

$$P(y_1|do(S = s)) = \int \int P(y_1|y_2, y_3)P(y_2|s)P(y_3|s)\, dy_2 dy_3$$
$$= \int \int P(y_1|y_2, y_3)P(y_2, y_3|s)\, dy_2 dy_3$$
$$= \int P(y_1|y_3)P(y_3|s)\, dy_3$$
$$= P(y_1|s).$$

This holds similarly for the configurations in Fig. 8A-B where the brain zone nodes are permuted.

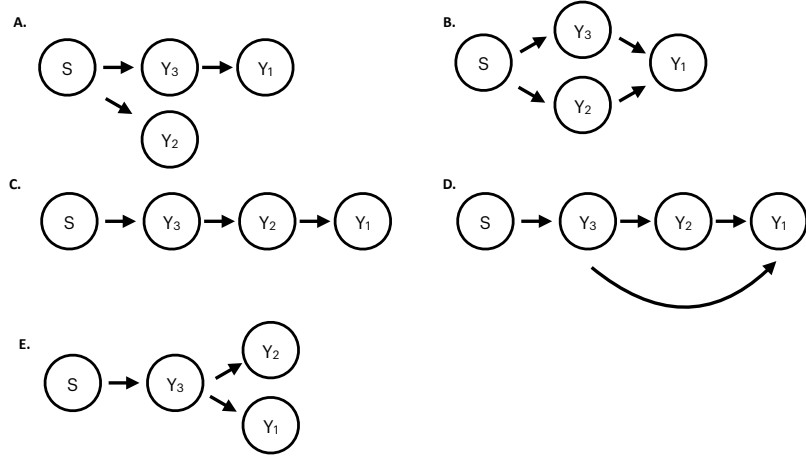

Figure 8: Causal DAG configurations for three brain zones and a stimulus. Under our paradigm and assumptions, these are all the possible configurations, where the stimulus $S$ affects both $Y_1$ and $Y_2$ and the third brain zone $Y_3$ has a causal relationship with $Y_1$ and/or $Y_2$, up to permutations of the brain zones.

Next, we consider all of the configurations where the third brain zone ($Y_3$) has a causal relationship with both $Y_1$ and $Y_2$ (Fig. 8C-E), up to permutations of the brain zones). We show that $P(y_i|do(S = s)) = P(y_i|s)$. In Fig. 8C-E), for $Y_3$ we can again use the truncated factorization to show that $P(y_3|do(S = s)) = P(y_3|s)$. For Fig. 8C $Y_2$, Fig. 8D $Y_2$, Fig 8E $Y_2$ and $Y_1$, we can apply the same

reasoning that applied to $Y_1$ in Fig. 7 to show that $P(y_i|do(S = s)) = P(y_i|s)$. For the last variable in Fig. 8C $Y_1$ we can use the truncated factorization and marginalization over $Y_2$ and $Y_3$ to show that:

$$P(y_1|do(S = s)) = \int \int P(y_1|y_2)P(y_2|y_3)P(y_3|s) \, dy_2 dy_3$$

$$= \int P(y_1|y_3)P(y_3|s) \, dy_3$$

$$= P(y_1|s).$$

For the last variable in Fig. 8D $Y_1$ we can use the truncated factorization and marginalization to show that:

$$P(y_1|do(S = s)) = \int \int P(y_1|y_2, y_3)P(y_2|y_3)P(y_3|s) \, dy_2 dy_3$$

$$= \int P(y_1|y_3)P(y_3|s) \, dy_3$$

$$= P(y_1|s).$$

$P(y_i|do(S = s)) = P(y_i|s)$ holds similarly for the configurations in Fig. 8C-E in the case where the brain zone nodes are permuted.

We have illustrated that for both the two and three brain zone settings that we can estimate the causal effect of the stimulus on each brain zone as $P(y_i|do(S = s)) = P(y_i|s)$.

## Appendix B. Metric Normalization

The main metrics of interest defined in Section 3 are encoding model performance, zone generalization, and zone residuals. In the simple setting where all annotated regions in Fig. 9 are independent of each other, the encoding model performance is proportional to annotated regions **1 + 2**, zone generalization to region **1**, and zone residuals to regions **2 + 3** (and to the analogous regions **2 + 3** in the other brain zone). For some scientific questions, it may be more informative to normalize these metrics in different ways. For example, one may normalize the zone generalization for a target brain zone by the encoding model performance for the same brain zone to compute the proportion of $\frac{1}{1+2}$ (i.e. the proportion of information shared between a target brain zone and the stimulus-representation that is also shared by a second brain zone). This metric is identical to the one proposed by Toneva et al. (2020). Another type of normalization that we find informative in the current work is the inter-subject correlation (ISC), which is proportional to **1 + 2 + 3 + 4** (i.e. the information shared between a target brain zone and the stimulus). This metric can be thought of as an estimate of the maximum possible performance (i.e. the noise ceiling). A similar metric was used as an estimate of the noise ceiling by Wehbe et al. (2021), though the authors did not make the connection to ISC explicitly. Note that the ISC across a dataset of more than two subjects is most frequently computed as the average of the pairwise ISC (i.e. the ISC for 1 of 6 subjects is the average across the ISC computed between that subject and the remaining 5 subjects). Following previous work (Hsu et al., 2004; Lescroart and Gallant, 2019), we normalize all of our metrics by the square-root of the noise ceiling, yielding normalized correlation values.

In our experiments, we specifically chose to normalize the zone generalization of zone 1 to zone 2 by the ISC for zone 2. We chose this normalization for two main reasons. Firstly, ISC

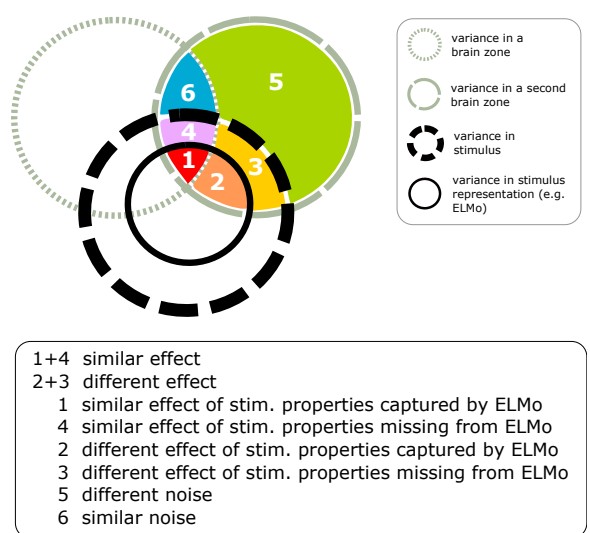

Figure 9: An illustration of possible effects that a stimulus can have on two brain zones, and the relationships with a stimulus-representation. Annotated regions **1 + 4** correspond to the stimulus having a similar effect on both brain zones, while **2 + 3** correspond to the stimulus having a different effect on the two brain zones (i.e. the stimulus overlaps with only one brain zone, not both brain zones). **1 + 2** indicate that the stimulus-representation captures the stimulus properties that cause at least part of the effect in the brain zone (**1** indicates the properties that have a similar effect on both brain zones, and **2** indicates the properties that have a different effect on both brain zones). Similarly, **3 + 4** indicate that the stimulus-representation is missing some stimulus properties that cause the specific effect in the brain zone. **6** indicates the shared noise between the brain zones, that is unrelated to the stimulus, and **5** indicates the noise that is unique to each brain zone.

can be thought of as an estimate of the maximum possible performance as we discussed above. That's not the case for encoding model performance. For example, in Fig. 11 (2nd and 3rd rows, rightmost cells), we see that if the target brain zone 2 has higher signal-to-noise ratio (SNR) than the source brain zone 1 (i.e. $\beta_2 > \beta_1$), then `zone generalization`$(zone_1, zone_2) >$ `encoding model performance`$(zone_1)$ for some levels of similarity of response to stimulus properties ($\alpha = 0.75$ and $\alpha = 1$). Therefore, the encoding model performance in the source zone is not an upper bound on the zone generalization. Secondly, we normalize by ISC specifically for the target zone 2, because we find in simulations that the SNR in the target zone is much more of a limiting factor to zone generalization (Fig. 11, 3rd row) than the SNR in the source zone.

We hope that the conceptual breakdown that we present in Fig. 9 will help other researchers choose the most relevant normalization for their questions of interest.

One additional consideration is that most metric normalization approaches rely on data which is in a shared anatomical space. It is common to analyze neuroimages in shared anatomical space, and we build on prior neuroimaging analyses and calculated our main metrics of interest and metric normalization (i.e., ISC) on the 268 ROIs defined by the Shen atlas in MNI space (Finn et al., 2015; Shen et al., 2013). However, the alignment of brain data to template space may be inexact, and atlas-defined brain regions may suffer from topological variability (i.e., variability in functional-anatomical correspondence) between participants (Salehi et al., 2020; Yaakub et al., 2020; Bohland

et al., 2009). A future extension of our work can address this limitation by not using standardized space or atlas-defined ROIs and instead rely on approach such as shared response models (Chen et al., 2015), regularized correlation analysis (Bilenko and Gallant, 2016), or hyperalignment (Haxby et al., 2011), which can be adapted to estimate a shared space from participants with different native spaces. Since our observations are relatively consistent between two real fMRI datasets containing two unique sets of participants, we do not expect that this limitation will drastically change the crux of our findings, but it might lead to an improved ability to infer the relationship between brain zones.

## Appendix C. Simulations

### C.1. Data Generation Model

**Simulating stimulus information.** We generate two components that together make up all available stimulus information: $X \in \mathbb{R}^d$, which is the stimulus-representation, and $Z \in \mathbb{R}^d$, a representation of the remaining stimulus information that $X$ does not capture. This is done by decomposing each of $X$ and $Z$ into four disjoint independent subsets of stimulus information: unique information that the individual brain zones respond to $(X_1, X_2, Z_1, Z_2)$, joint information that both brain zones respond to $(X_{12}, Z_{12})$ and information that neither brain zone responds to $(X_3, Z_3)$. Each $X_i, Z_i$, of length $\frac{d}{4}$, is independently sampled from a multivariate normal with mean 0 and a symmetric toeplitz covariance matrix with diagonal elements equal to 1. $X$ and $Z$ are then constructed by concatenating their four corresponding sub-components.

**Simulating brain zone data.** We simulate observations at two brain zones from two distinct participants using the following data generation model (motivated by Eq. 2):

$$Y_{i,P} = \alpha \times \underbrace{g_{12,P}(X)}_{\text{joint signal}} + (1 - \alpha) \times \underbrace{g_{i,P}(X)}_{\text{unique signal}} + \alpha \times \underbrace{N_{i,P}}_{\text{unique noise}} + (1 - \alpha) \times \underbrace{N_{12,P}}_{\text{joint noise}} \quad (6)$$

where $N_{i,P} = \delta \times h_{i,P}(Z) + (1 - \delta) \times \epsilon_{i,P}$ and $N_{12,P} = \delta \times h_{12,P}(Z) + (1 - \delta) \times \epsilon_{12,P}$.

Here, each $g_{i,P}(X) = \langle \theta_{i,P}, X_i \rangle$ is a linear function of the stimulus-representation that selectively acts on the corresponding $X_i$ in $X$. In order to generate the necessary participant-specific parameters $\theta_{i,P} \in \mathbb{R}^{\frac{d}{4}}$, we first generate $\theta_i \in \mathbb{R}^{\frac{d}{4}}$ by independently sampling each of its components from a uniform distribution over $[0, 1)$. Each $\theta_{i,P}$ is then sampled from $\mathcal{N}(\theta_i, 0.25\mathbf{I})$ to allow for variation between participants. The same approach is used to generate each $h_{i,P}(Z) = \langle \phi_{i,P}, Z_i \rangle$ term. $\epsilon_1, \epsilon_2, \epsilon_{12} \in \mathbb{R}$ are terms that represent the information captured that is not related to the stimulus. Each $\epsilon_i$ is independently sampled from a standard normal distribution. Finally, we use the standardized values of each of the signal and noise components in our data generation model (Eq. 6), as this allows us to preserve the variance of the overall signal to noise. For simplicity, we omit this detail from Eq. 6.

In the data generation model above, we introduced two adjustable parameters $\alpha$ and $\delta$ to simulate a wide range of scenarios. Parameter $\alpha \in [0, 1]$ controls how similarly the two zones respond to the stimulus properties captured in the stimulus-representation. We designed the weightings in Eq. 6 such that the total variance of each of the following four components remains constant when varying $\alpha$: the total signal (joint+unique), noise (joint+unique), joint information (signal+noise) and unique information (signal+noise). Parameter $\delta \in [0, 1]$ controls the proportion of stimulus properties that are driving the brain zones but are not captured by the stimulus-representation. The results shown in Fig. 3(Left) and Fig. 3(Right) were collected by varying $\alpha$ (when $\delta = 1.0$) and $\delta$

(when $\alpha = 1.0$) respectively in the simulations that were performed. This allowed us to smoothly interpolate between four inferences that we want to, but cannot, distinguish between using just encoding model performance.

### C.2. Additional Simulation Results

In Fig. 10, we present how encoding model performance, zone generalization, zone residuals, and functional connectivity vary as we allow $\alpha, \delta \in [0, 1]$ to vary with respect to each other in Eq. 6. Since inferences A and B are characterized by both zones responding differently to the stimulus properties captured in the stimulus-representation and inferences C and D are characterized by both zones responding similarly to them, increasing $\alpha$ from 0.0 to 1.0 lets us adjust from the former pair of inferences to the latter. Also recall that one can separate inference A from B and inference C from D, if one has information about the extent to which both zones respond to stimulus properties not captured by the stimulus-representation. Therefore, at a high fixed $\alpha$, as $\delta$ is increased from 0.0 to 1.0, we move from inference B to inference A or inference C to inference D (depending on the pair we narrowed down earlier).

Fig. 10 shows that zone generalization is the only metric of the four we consider that can be used to separate inferences A and B from inferences C and D - i.e., distinguish between brain zone data that is simulated using low and high values of $\alpha$ respectively at any choice of fixed $\delta$. This aligns with our observations from Fig. 3(Left), where we concluded that looking at zone generalization can help us identify which pair of inferences we should further investigate. To know what we can precisely infer, we would need a metric that lets us distinguish between brain zones based on the extent to which they respond to stimulus properties that are not captured by the stimulus-representation - in our simulations, between brain zone data simulated at different values of $\delta$ when $\alpha$ is high. Fig. 10 shows that at high $\alpha$, zone residuals increases with $\delta$, and therefore, can be useful when separating inferences B and C from inferences A and D. This also aligns with our observations from Fig. 3(Right).

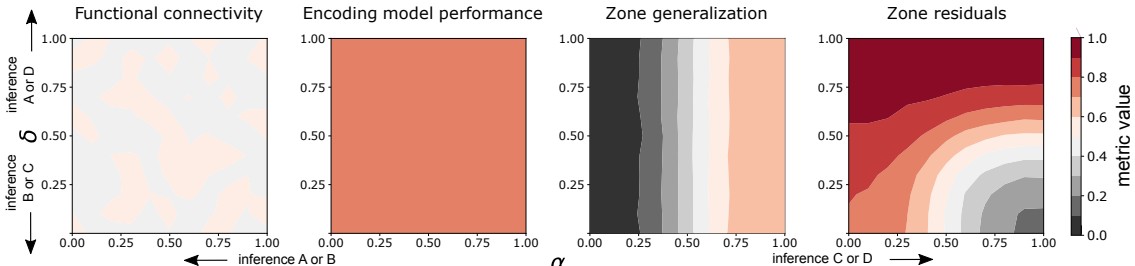

Figure 10: How each metric varies under simulations performed at different settings of $\alpha, \delta$.

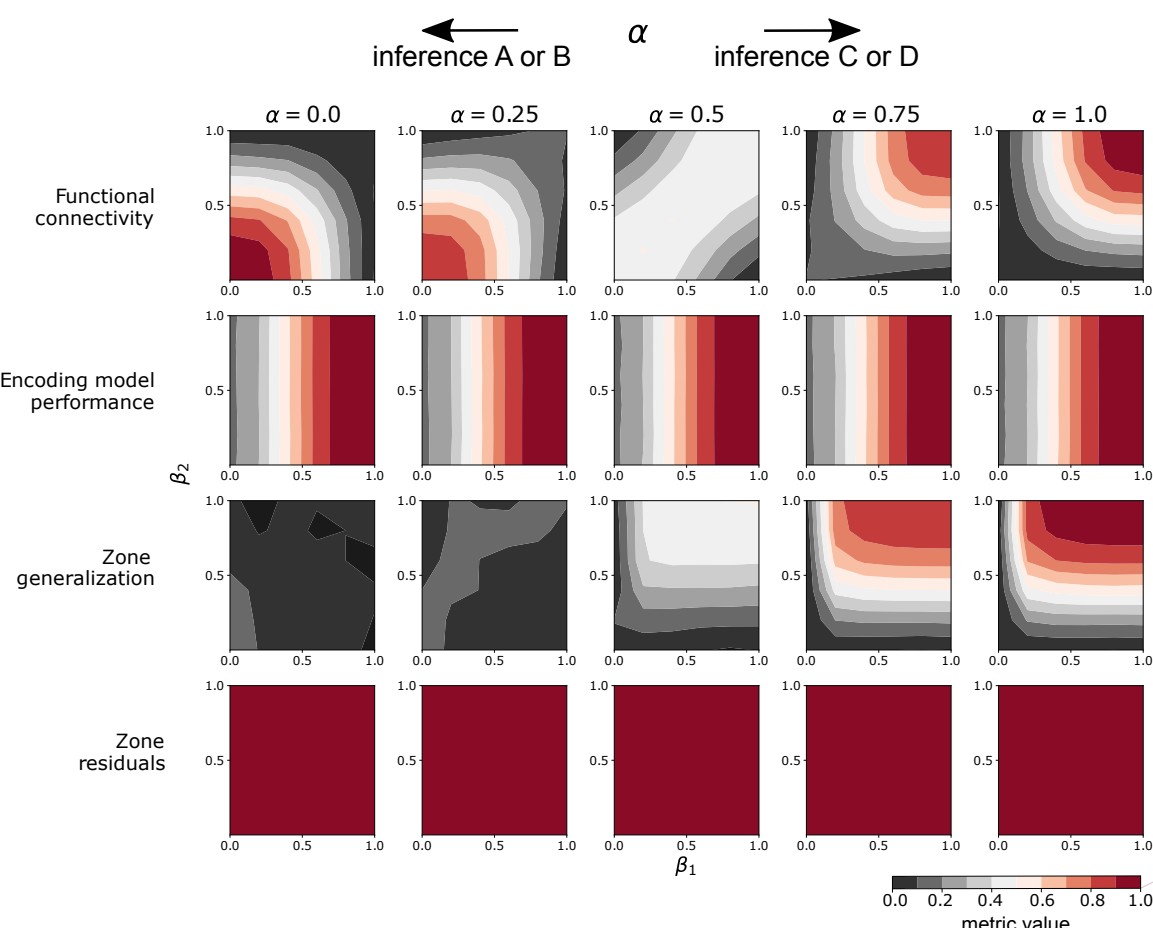

Figure 11: Extending Fig. 3(Left), this figure shows how each metric varies under simulations performed at different signal-to-noise ratios as we vary $\alpha, \beta_1, \beta_2$ when $\delta = 1.0$ is fixed.

### C.3. Varying signal-to-noise ratio in simulated brain zone data

To test the limits of what each metric can tell us about the underlying relationships between a pair of brain zones, we further extend Eq. 6 to control the extent to which the activity in each zone is driven by aspects of the stimulus that are captured by (vs. missing from) the stimulus-representation:

$$Y_{i,P} = \beta_i \underbrace{[\alpha \times g_{12,P}(X) + (1 - \alpha) \times g_{i,P}(X)]}_{\text{signal}} + (1 - \beta_i) \underbrace{[\alpha \times N_{i,P} + (1 - \alpha) \times N_{12,P}]}_{\text{noise}} \quad (7)$$

$$\text{where } N_{i,P} = \delta \times h_{i,P}(Z) + (1 - \delta) \times \epsilon_{i,P} \text{ and } N_{12,P} = \delta \times h_{12,P}(Z) + (1 - \delta) \times \epsilon_{12,P}.$$

Above, we introduce an additional type of parameter $\beta_i \in [0, 1]$. In this context, zone activity that is driven by stimulus properties captured by the stimulus-representation can be viewed as the signal that is retrievable by an encoding model. The remaining activity, whether stimulus-driven or not, can be viewed as the noise that an encoding model cannot explain as it only has access to the stimulus-representation. Additionally, we use the standardized values of the signal and noise components to preserve the variance of the overall signal to noise in Eq. 7, similarly to in Eq. 6.

First, we focus on the separation between the pairs of inferences A and B and inferences C and D by varying $\alpha$. In Fig. 11, we show how each metric varies as we vary $\beta_1$ and $\beta_2$ for each setting of $\alpha$. $\beta_1$ and $\beta_2$ control the signal-to-noise ratio in both simulated brain zones. We fix $\delta = 1.0$ here, but similar trends can be observed for other choices of fixed $\delta \in [0, 1]$ as well. We observe that encoding model performance and zone residuals do not allow us to distinguish between different $\alpha$ values. Note also that functional connectivity cannot be used to identify when both brain zones mostly respond to shared noise (low $\beta_i$'s, low $\alpha$) from when they mostly respond to shared signal (high $\beta_i$'s, high $\alpha$). We observe that given sufficient signal in both brain zones ($\beta_1, \beta_2 \gg 0$), zone generalization increases as $\alpha$ increases. However, under conditions of little to no signal, zone generalization cannot be used to distinguish between different $\alpha$ values. These results suggest that zone generalization is a useful metric to separate the pair of inferences A and B (low $\alpha$) from the pair C and D (high $\alpha$) only when the encoding models used are able to perform relatively well on the brain zones they are trained on.

Next, we consider the separation between the pairs of inferences B and C and inferences A and D by keeping a high fixed $\alpha$ and varying $\delta$. In Fig. 12, we show how each metric varies as we vary $\beta_1$ and $\beta_2$ for each setting of $\delta$ (when $\alpha = 1.0$). We find that encoding model performance, zone generalization and functional connectivity are not useful to distinguish between different values of $\delta$ in our simulations. We observe that given sufficient noise in brain zone 1 ($\beta_1 \ll 1$), zone residuals increases as $\delta$ increases. This suggests that the zone residuals can help us separate inferences B and C (low $\delta$) from inferences A and D (high $\delta$). However, in the case when $\beta_1$ is high, increasing $\delta$ does not impact the zone residuals as they are already saturated. In the case where the zone residuals are already saturated, the zone residuals enable us narrow our search down to inferences A or D.

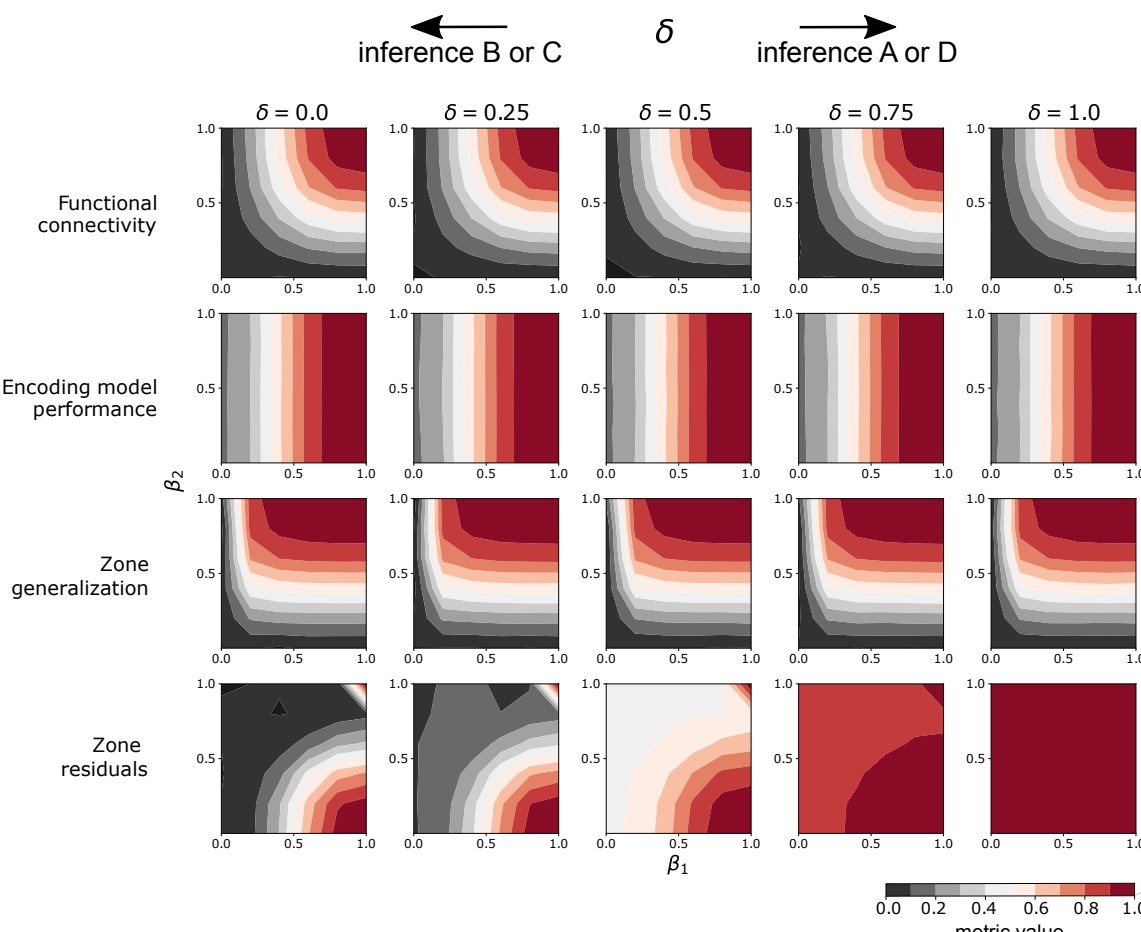

Figure 12: Extending Fig. 3(Right), this figure shows how each metric varies under simulations performed at different signal-to-noise ratios as we vary $\delta, \beta_1, \beta_2$ when $\alpha = 1.0$ is fixed.

### C.4. Simulations of Alternate Data Generation Models

We evaluate if our proposed framework can help us infer if a stimulus affects two brain zones in the same way for mildly misspecified alternate data generation models. We introduce two alternate data generation models, where $Y_1$ is the cause of $Y_2$, then we evaluate the three metrics in our framework on data synthetically generated from these data generation models.

In the first alternate model, $X$ (the stimulus-representation) and $Z$ (the remaining stimulus information not captured by $X$) directly affect $Y_2$ and indirectly affect $Y_2$ through mediator $Y_1$:

$$Y_{1,P} = g_{12,P}(X) + \delta \times h_{12,P}(Z) + (1 - \delta) \times \epsilon_{12,P} \tag{8}$$

$$Y_{2,P} = \tau \times Y_{1,P} + g_{2,P}(X) + \delta \times h_{2,P}(Z) + (1 - \delta) \times \epsilon_{2,P} \tag{9}$$

In the data generation model above, we introduced an adjustable parameter $\tau$ to simulate a wide range of scenarios. Parameter $\tau \in [-1, 1]$ controls the extent and direction that $Y_2$'s measurements are affected by $Y_1$. The remaining parameters and functions were previously defined in Appendix C.1.

In the second data generation model, $X$ and $Z$ only indirectly affect $Y_2$ through mediator $Y_1$:

$$Y_{1,P} = g_{12,P}(X) + \delta \times h_{12,P}(Z) + (1 - \delta) \times \epsilon_{12,P} \tag{10}$$

$$Y_{2,P} = \tau \times Y_{1,P} + \epsilon_{2,P} \tag{11}$$

We evaluate the three metrics in our framework on synthetic data from both data generation models in Figs. 13, 14. We find that only zone generalization is informative of how similar two brain zones respond to stimulus properties, enabling us to determine which pair of inferences (A or B vs. C or D) can be made (Figs. 13, 14(Left)). We also find that for the first data generation model only zone residuals vary when the proportion of stimulus properties that are driving the activity in the brain zones but not captured by the stimulus-representation changes (Fig. 13(Right)). For the second data generation model, we find that when we vary the proportion of stimulus properties that are driving the activity in the brain zones but not captured by the stimulus-representation the zone residuals remain constant at (Fig. 14(Right)). This is expected as the stimulus properties that drive the brain zones are shared between the two zones. Therefore, for the second data generation model only inference B or C is possible. This suggests that for both models, we can use zone generalizations to narrow our search down to a pair of inferences (A or B vs. C or D), and then zone residuals can be used to precisely infer how the stimulus affects the pair of zones. Therefore, even for these mildly misspecified data generation models when used together zone generalization and zone residuals enable us to infer if a stimulus affects two brain zones with significant encoding model performance in the same way.

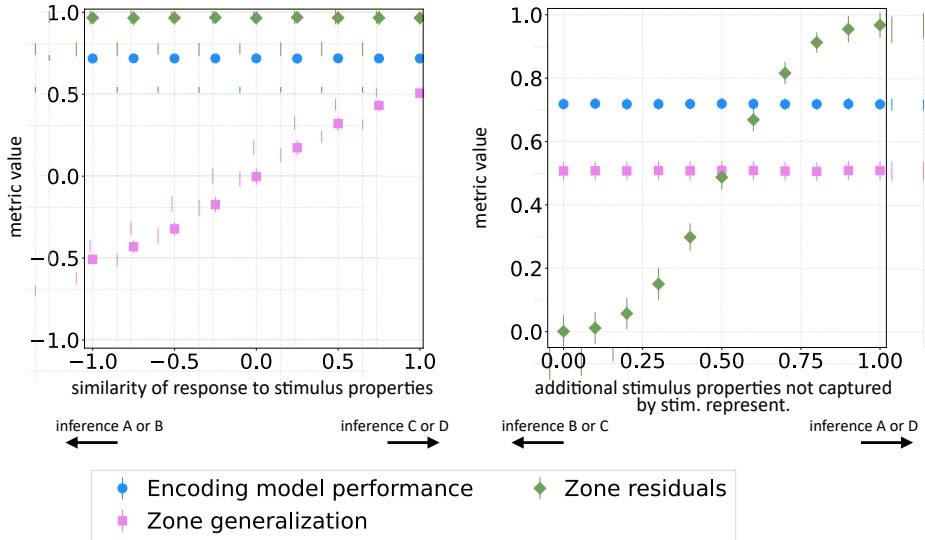

Figure 13: First alternate data generation model. Average metric values under simulations that separate (Left) inference A or B from inference C or D and (Right) inference B or C from inference A or D.

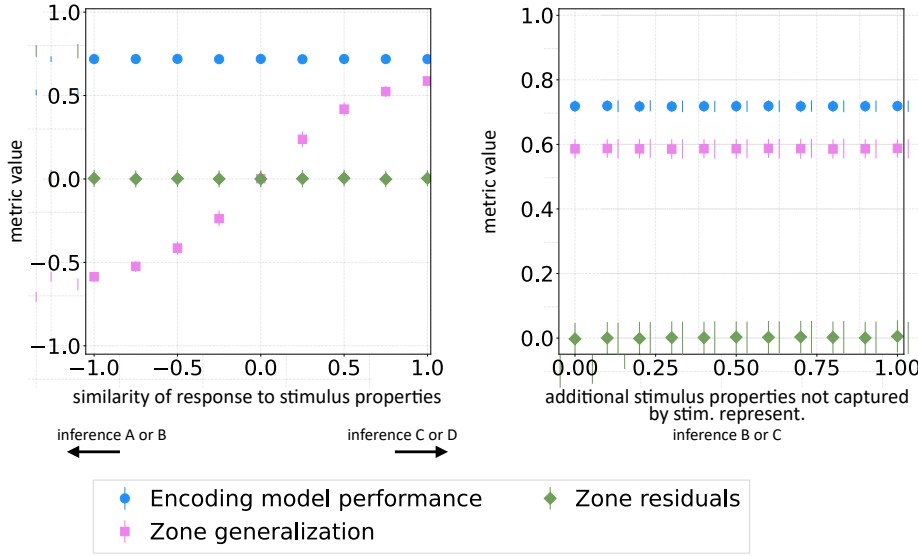

Figure 14: Second alternate data generation model. Average metric values under simulations that separate (Left) inference A or B from inference C or D and (Right) show that zone residuals remain small and therefore the inference must be inference B or C.

### C.5. RSA and Functional Connectivity Simulation Analyses

We also tested whether RSA or functional connectivity are able to infer the true underlying relationships in the same synthetic brain zone dataset described in Section 4. In Fig. 15 we extended Fig. 3 to include these RSA and functional connectivity metrics. We conducted two types of RSA: (1) between an RDM corresponding to each of the brain zones and an RDM corresponding to the synthetic stimulus-representation, and (2) between the two brain zone RDMs. What we found was that, similarly to encoding model performance in Fig. 3, all of the RSAs resulted in a flat line as we varied (1) how similarly the zones respond to the stimulus properties captured by the stimulus-representation (i.e. different values of $\alpha$) (Fig. 15(Left)) and (2) the extent to which both zones respond to stimulus properties not captured by the stimulus-representation (i.e. different values of $\delta$) (Fig. 15(Right)). The same is true of functional connectivity.

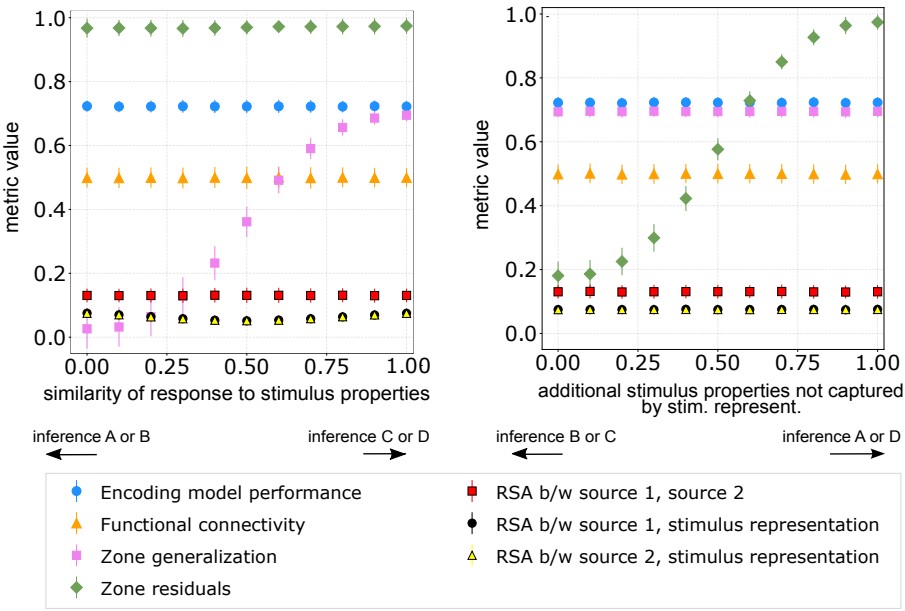

Figure 15: Extending Fig. 3, this figure shows how different RSA-based metrics and functional connectivity vary on the same synthetic dataset.

## Appendix D. Data Preprocessing

### D.1. HCP

Our analyses are performed with the 3105 TRs (51 minutes and 45 seconds) suggested for analysis in the HCP documentation. These exclude rest periods and the first 6 TRs of each movie clip within a movie run. Individual-level results are presented for the six participants with the highest encoding model performance averaged over the 55 Shen atlas language ROIs.

### D.2. Courtois NeuroMod

Results included in this manuscript come from preprocessing performed using fMRIPrep 20.1.0 (Esteban et al., 2018b,a). Three participants are native French speakers and three are native English speakers. All participants are fluent in English and report regularly watching movies in English.

### D.3. Other Pre-processing

The fMRI datasets and Shen atlas were provided in different template spaces and voxel sizes. We resample and register the Shen atlas (MNI27 template space, voxel size = 1 mm isotropic) to both the HCP template space (MNI152NLin6Asym, voxel size = 1.6 mm isotropic) and the Courtois NeuroMod template space (ICBM2009cNlinAsym, voxel size = 2 mm isotropic) using FSL FMRIB Linear Image Registration Tool (FLIRT) (Jenkinson et al., 2002). We perform all analyses for the two datasets in their respective template space.

We further process the ELMo embeddings before we use them as the input features to our encoding models. First, we use a Lanczos filter with the same parameters as Huth et al. (2016) to downsample the embeddings into a feature matrix where each row corresponds to a feature vector for a TR. Then, to reduce the dimensionality of our feature space we use principle component analysis (PCA) to select the first 10 principle components. The first 10 principle components explain 50.5% of the variance in the Courtois NeuroMod dataset and 49.9% of the variance in the HCP dataset. Next, to account for the lag in the hemodynamic response in fMRI data, we delay the feature matrix in accordance with previous work (Nishimoto et al., 2011; Wehbe et al., 2014a; Huth et al., 2016).

## Appendix E. Negative Normalized Zone Generalization

We investigated why negative norm. zone generalizations arise. We have found in our empirical results that when there is a negative norm. zone generalization, the two ROIs have different positive weights on the features in the stimulus-representation. For example, some ROIs put high weights on features associated with word rate, while other ROIs put more weight on the rest of the stimulus-representation. This suggests that at least some stimulus properties affect the two ROIs differently. However, it is unclear what these ROIs respond to besides word rate. The negative norm. zone generalization suggests that the ROIs have opposite weights on the features in the stimulus-representation. This suggests that at least some stimulus properties affect the ROIs differently, however it is unclear if the stimulus-representation is incomplete and does not include all the stimulus properties that affect the ROIs similarly. Consequently, from a negative zone generalization alone it is not possible to infer if the stimulus properties affect the ROIs mostly differently (inference A) or if some stimulus properties affect ROIs differently and other properties not captured by ELMo affect the ROIs similarly (inference B). Therefore, to interpret the negative norm. zone generalization values we also need the zone residuals.

## Appendix F. Stability of Zone Residuals With Increasing Numbers of Participants

As the zone residuals metric takes into account each possible pair of participants, we evaluated the stability of the zone residuals metric with increasing numbers of participants. We evaluated the stability by calculating the zone residuals metric for 88 sample sizes (ranging from 2 to 89 participants) from the HCP dataset. For each sample size $n$ we randomly sampled $n$ participants

100 times, each time sampling without replacement. Then we calculated the zone residuals between each pair of brain zones. As it is difficult to visualize the results for all ROI pairs we present the average zone residual value for each sample size for the ROI pairs shown in Fig. 6 (Fig. 16). The zone residuals' stability increases with increasing sample size until it stabilizes with a sample size of 5-10 participants. Therefore, we recommend using zone residuals for datasets with at least five participants.

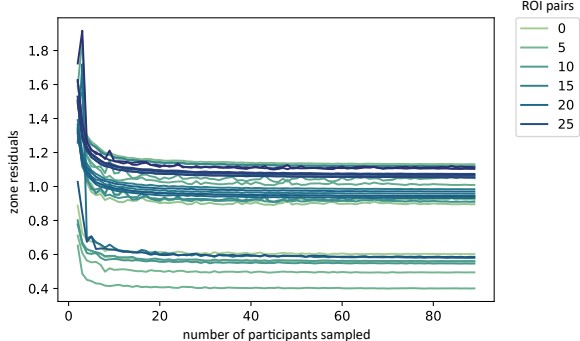

Figure 16: The impact of sample size on zone residuals metric. The zone residuals metric stability increases with increasing sample size until it stabilizes with a sample size of 5-10 participants. The ROI pairs presented are those shown in Fig. 6.

## Appendix G.  Asymmetry in Zone Generalization and Zone Residuals

We observe that many pairs of brain zones exhibit asymmetric zone generalization and zone residual values. The asymmetry in both zone generalization and zone residuals can be due to a difference in signal-to-noise ratio (SNR) between the brain zones, and also to a difference in the proportions of all stimulus properties that similarly affect the zones. To illustrate this, consider the extreme case in which the stimulus properties that affect zone 1 are a strict subset of those that affect zone 2 and the effect on the two zones is similar. Then,

$$\texttt{zone generalization}(zone_1, zone_2) < \texttt{zone generalization}(zone_2, zone_1),$$
$$\texttt{zone residuals}(zone_1, zone_2) < \texttt{zone residuals}(zone_2, zone_1).$$

Disentangling this from the SNR effect on asymmetry is an interesting direction for future work.

## Appendix H.  34 Language ROI Heatmap Tick Numbers Projected on the Brain

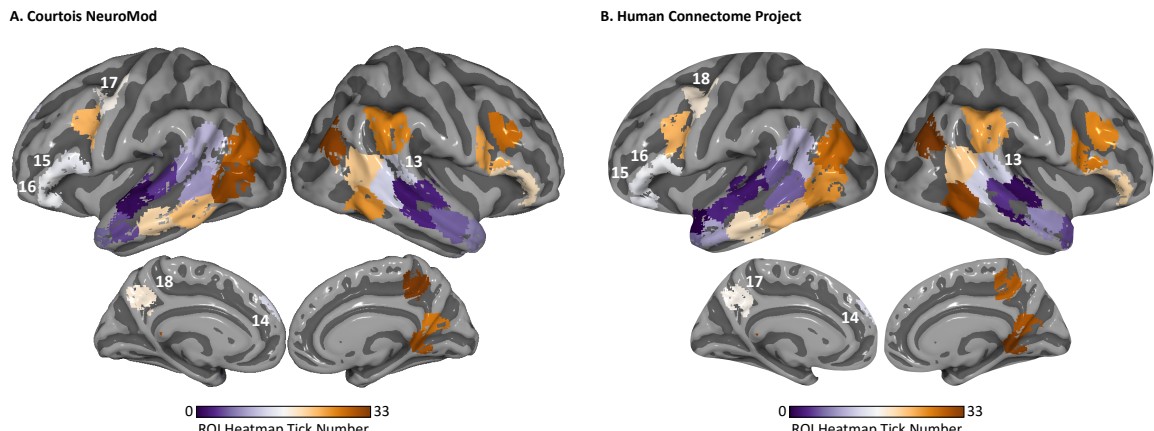

Figure 17: ROI Heatmap Tick Numbers. The 34 significant language ROIs are shown on the cortical surface and colored according to the tick number for each ROI in the (A) Courtois NeuroMod and (B) HCP heatmaps in Figs. 5, 18, 20, 21. The ROIs are sorted from high (ROI 0) to low (ROI 33) median normalized zone generalization (average over participants).

## Appendix I. Normalized Zone Residuals on Two Naturalistic fMRI Datasets

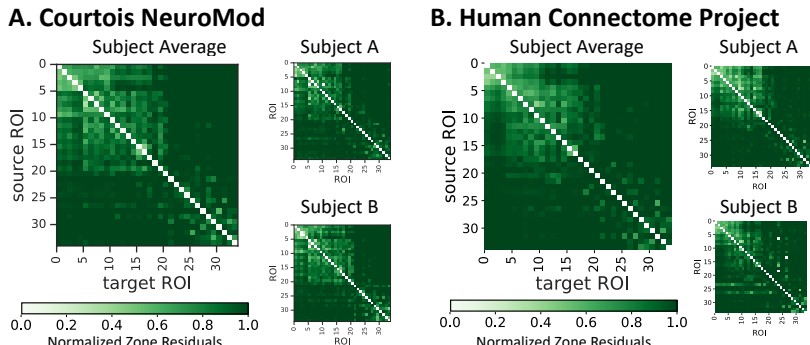

Figure 18: Zone Residuals. ROI pairs with large norm. zone residuals (dark green) are affected differently by at least some stimulus properties (inference A or D). These ROI pairs with large norm. zone residuals are consistent at the group and participant-level in both datasets.

## Appendix J. Additional Participant-Level Empirical Results

We present the participant-level results for the remaining four participants in the Courtois NeuroMod dataset and four additional participants for the HCP dataset. We observe that these additional participants appear similar to the average and two representative participants presented in the main text and Appendix I.

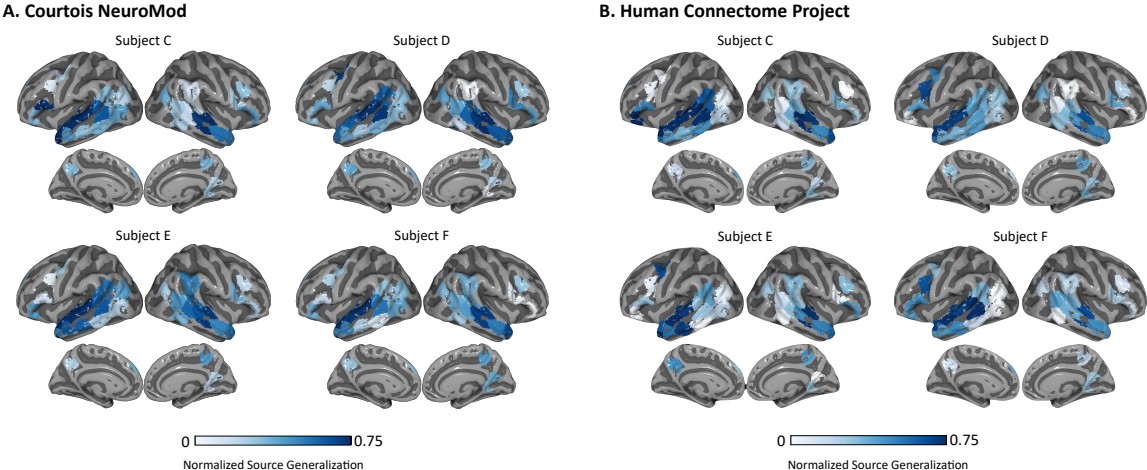

Figure 19: (Related to Fig. 4) Encoding Model Performance. Similar to Fig. 4 in the main text, this figure shows the normalized encoding model performance at 34 significantly predicted ROIs (corrected at level 0.05) for participants C-F in both the (A) Courtois NeuroMod and (B) Human Connectome Project datasets. Plots were created using the Pycortex software (Gao et al., 2015).

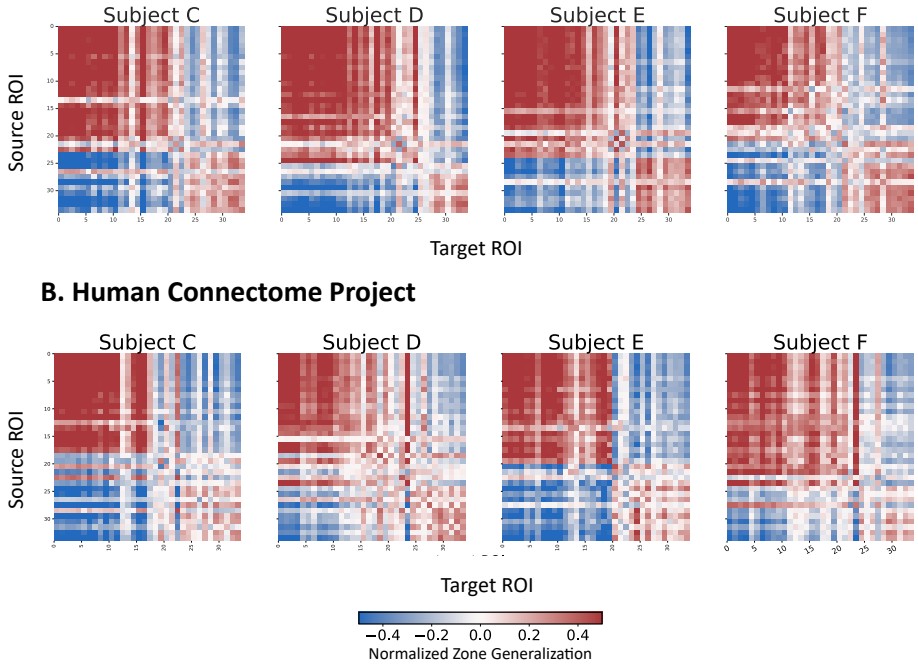

Figure 20: (Related to Fig. 5) Zone Generalization. Similar to Fig. 5 in the main text, this figure shows the normalized zone generalization for participants C-F in both the (A) Courtois NeuroMod and (B) Human Connectome Project datasets. ROI pairs with high normalized zone generalization (red) are consistent across participants C-F in both datasets. They are also consistent with the group level and participants presented in the main text.

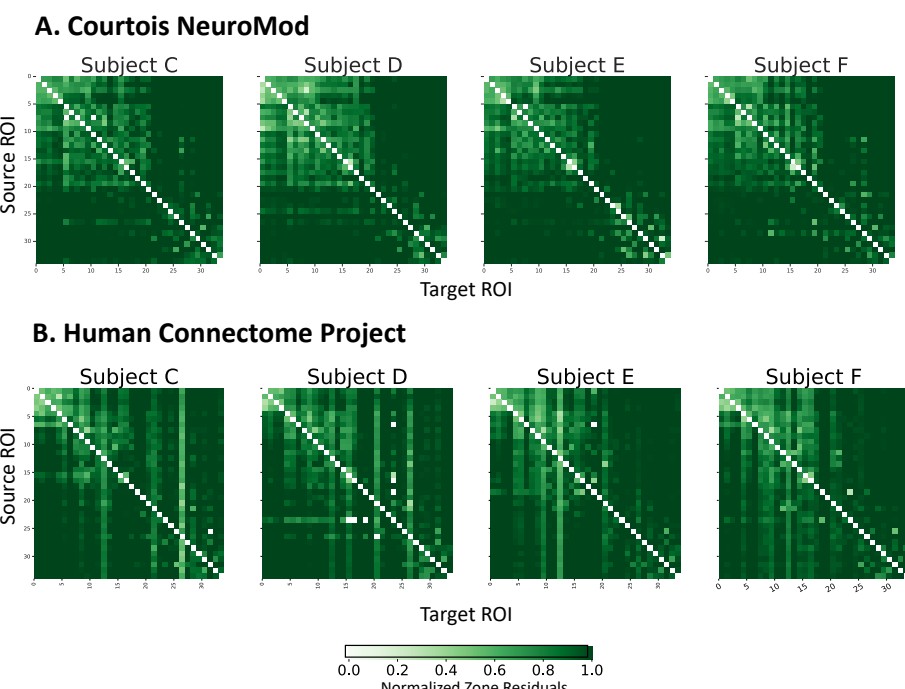

Figure 21: (Related to Appendix Fig. 18) Zone Residuals. Similar to Appendix Fig. 18, this figure shows the normalized zone residuals for participants C-F in both the (A) Courtois NeuroMod and (B) Human Connectome Project datasets. ROI pairs with high normalized zone residuals (dark green) are consistent across participants C-F in both datasets. They are also consistent with the group level and participants presented in Appendix Fig. 18.

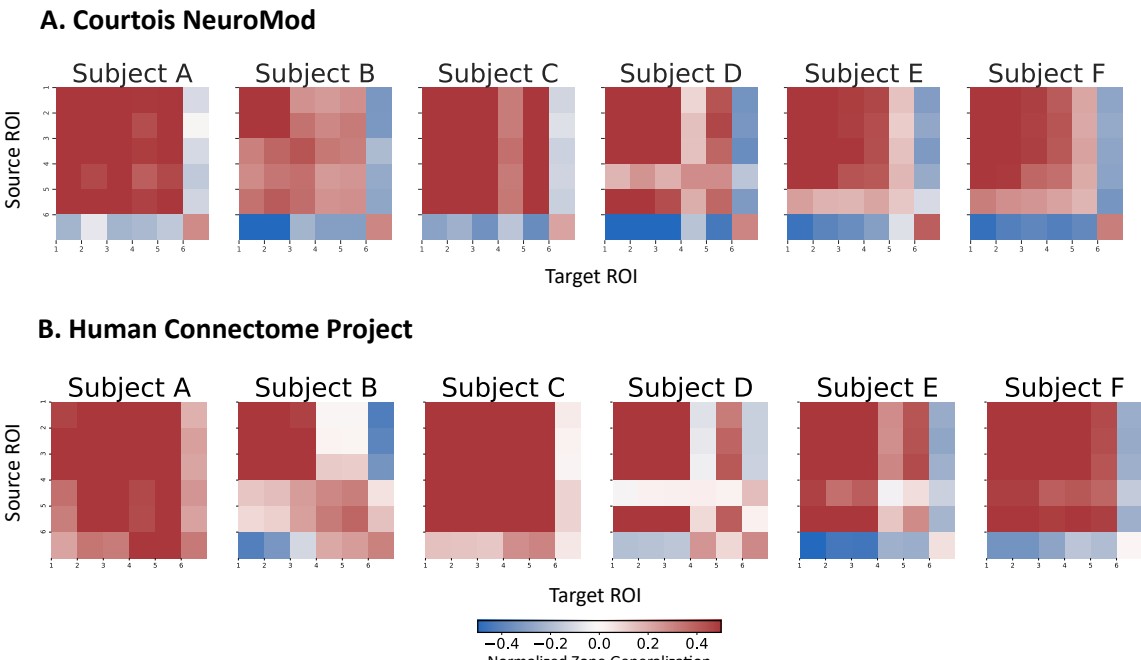

Figure 22: (Related to Fig. 6) Proposed Framework Example Participant-Level Zone Generalization. Similar to Fig. 6 in the main text, this figure shows the normalized zone generalization for the six ROIs in the example using the proposed framework for participants A-F in both the (A) Courtois Neuromod and (B) Human Connectome Project datasets. The ROI pairs with high normalized zone generalization (red) are consistent across participants A-F in both datasets. They are also consistent with the group level presented in the main text.

Figure 23: (Related to Fig. 6) Proposed Framework Example Participant-Level Zone Residuals. Similar to Fig. 6 in the main text, this figure shows the normalized zone residuals for the six ROIs in the example using the proposed framework for participants A-F in both the (A) Courtois Neuromod and (B) Human Connectome Project datasets. The ROI pairs with high normalized zone residuals (dark green) are consistent across participants A-F in both datasets. They are also consistent with the group level presented in the main text.

