# OpenReview forum: "Same Cause; Different Effects in the Brain"
_cclear.cc/CLeaR/2022/Conference — CLeaR 2022 Poster_

### Official Review · Reviewer_czAT · 2021-11-10

**Confidence:** 3
**Overall Score:** 6

**Main Review:**

The paper proposes an intriguing question, to study if two causal effects indeed share a common stimulus and if there are unobserved stimulus properties that can act as a common cause. The proposed metrics to answer the question are simple and easy to follow; the review below focuses only on the weaknesses of the paper.

Figure 1 is very helpful. Consider adding a Figure 0 that illustrates encoding model performance (i.e. Stimulus properties -> Stimulus representation via e.g., ELMo; recording activity in target brain zone; measuring predictability via "variance overlap" as in the first row of Fig 1).

Eqn 3: For the zone generalization metric, it seems important to normalize by the encoding model performance corr(hat{Y}_1, Y_1) and corr(hat{Y}_2, Y_2)? If not, how should we calibrate large vs. small zone generalization values corr(hat{Y}_1, Y_2)? The text also suggests that small values for zone generalization, *in addition to* good encoding model performance, tells us that two zones respond differently. Can we define the metric using relative correlation measures? Another naive question: Should zone generalization be a symmetric metric? I.e. combine corr(hat{Y}_1, Y_2) and corr(hat{Y}_2, Y_1) somehow?

For the regression in Eqn2, should the model still be fitted using the cross-validation procedure for fitting Eqn1 (for each zone in isolation)? Eqn2 suggests that we should fit the two encoding models jointly... Please clarify how hat{g}_1 and hat{g}_2 are fitted to training data.

Zone residuals: It will be helpful to give intuition for why zone residuals are not suitably computed for a single participant. Is there a fundamental non-identifiability of g and h in Eqn 4 if we only get to see a single participant? (The linear assumption on g likely avoids this non-identifiability). Is there non-identifiability between h and noise epsilon_i? Would structural assumptions on h help? How do multiple participants help to avoid this difficulty?

Again, Eqn4 suggests that we should fit the encoding models across all participants jointly. Please clarify how hat{g} and the residual regressions Y_1,P <- Y_2,P etc. are fitted to data.


Section 3 Simulations: All the simulations are well-specified/realizable. I.e., brain activity in each zone is indeed a function of X, Z and epsilon. It would be helpful to simulate mildly mis-specified settings: E.g. Zone 1 activity is a mediator or effect modifier for Zone 2 activity. (Does not fit the additive shared noise epsilon_12 model but could plausibly be true in actual fMRI data).
In general, Z being assumed as a common cause of Y_1 and Y_2 is a specific causal modeling assumption. It will be helpful to spell this out, and discuss alternate models for when zone generalization is large (e.g., that Y_1 is a cause of Y_2 or vice versa). Note that the zone residuals will have to be modified in the case of these alternate causal models. <-- This clarification around the modeling assumptions is the biggest opportunity to strengthen the paper.

Section 4: Naive question (connecting to the asymmetric nature of zone generalization and zone residual metrics) -- Do the asymmetric relationships inferred between different ROIs suggest a different underlying mechanism than a common cause Z (or is it fully explained by different base SNR for the different ROIs, contributing to an asymmetric zone generalization value)?

-----
(After author response)
I appreciate the authors' clarification to Q3 and Q8 (asymmetry in the metrics) and the response to Q7 (which was the weakest point) taken in conjunction with response Q4 to Nc86 have satisfied my concerns about the utility of these metrics in mis-specified settings.


**Summary:**

An intriguing question about causal effects in neuroscience, and tools from structural equation modeling to answer them.

---

> ### Author Response · Authors · 2021-12-04
> **Response to reviewer czAT, Part 1 of 2**
>
> We thank the reviewer for their thoughtful suggestions and believe that adding a figure that introduces encoding models will strengthen the paper. We will add such a figure to the updated manuscript. We attempt to address the reviewer's questions below.
> ### **Q1: normalization of zone generalization**
>
> **Summary:** We agree that normalization is important and we use normalization in all of the experiments, as discussed in Sec 2.4 and Appendix A. We did not include a specific normalization in the definition of zone generalization because different types of normalization may be suitable when using the metrics for investigating different scientific questions (Appendix A). We are happy to discuss the normalization earlier in Sec 2 if the reviewer would prefer that.
>
> **Further details:** In our experiments, we chose to normalize the zone generalization of zone_1 to zone_2 by the inter-subject correlation (ISC) for zone_2 (Sec 2.4). We chose this normalization for two main reasons. Firstly, ISC can be thought of as an estimate of the maximum possible performance (i.e. the noise ceiling, see Appendix A for a figure and a longer discussion). That’s not the case for encoding model performance. For example, in Fig 8 (2nd and 3rd rows, rightmost cells), we see that if the target brain zone_2 has higher SNR than the source brain zone_1 (beta_2 > beta_1), then zone generalization(zone_1, zone_2) > encoding model performance(zone_1) for some levels of similarity of response to stimulus properties (alpha=0.75 and alpha=1). Therefore, the encoding model performance in the source zone is not an upper bound on the zone generalization.
>
> Secondly, we normalize by ISC specifically for the target zone_2, because we find in simulations that the SNR in the target zone is much more of a limiting factor to zone generalization (Fig 8, 3rd row) than the SNR in the source zone.
>
> ###  **Q2: relative correlation measures**
>
> A relative correlation measure is not sufficient in all cases, because we want to distinguish the following two scenarios: 1) low encoding performance and low zone generalization, and 2) high encoding performance and high zone generalization. A relative correlation measure may make these two cases appear very similar, but we should not make any inference in the first case. One way to go around this is to threshold by the encoding model performance, which is equivalent to the first step in our framework (Fig 1, top).
>
> ### **Q3: symmetric zone generalization**
>
> We believe that the asymmetry of zone generalization can be informative (see response to **Q8** below). We agree with the reviewer that a symmetric version of zone generalization may also be informative, which can be explored by future work.
>
> ### **Q4: fitting of zone generalization to training data**
>
> **Summary:** We fit all three metrics based on the most general and most commonly-assumed model which treats each brain zone independently (encoding models and zone generalization are fit based on Eq1; zone residuals are fit based on Eq1 when the stimulus representation X is replaced by the brain recordings for a second brain zone Y_j).
>
> **Further details:** However, the true underlying model of how brain activity is generated as a function of the stimulus is of course unknown. In the remainder of Sec 2, we posit that there can be additional components in the brain activity, which are important for understanding whether two zones are affected by stimuli in a similar or different way. We introduce the two additional models (Eq2 & Eq4) to conceptually break down the independent component epsilon in Eq1 into these additional important components. Our simulations show that when these additional components are present in the brain data, our proposed metrics (zone generalization and zone residuals): 1) are indeed sensitive to these additional components, despite being fit based on the most general model in Eq 1, and 2) allow us to make one of the four main inferences in Fig 1, when used together. We will clarify this in the paper.
> In general, there can be other models that can be used to model brain zones jointly, but whether such approaches are advantageous is an open research question. For example, Wehbe et al. (AoAS 2015) showed that jointly modeling different voxels might help provide better estimates for noisy voxels, but that there is not much advantage for voxels that are already well modeled on their own.

---

> ### Author Response · Authors · 2021-12-04
> **Response to reviewer czAT, Part 2 of 2**
>
> Continued from Part 1
>
> ### **Q5: zone residuals and multiple participants**
>
> We resort to zone residuals because we want to account for features of the stimulus that we cannot capture in our stimulus representations and that are processed uniquely by one zone. We assume these features affect the brain zones reliably across participants, but we cannot access them. Thus we use the fact that the residual of one zone on another is correlated across participants as an estimate for the processing of these features in one of the zones and not the other.
> One other option is to correlate the residuals across repetitions of the same stimulus in the same participant (if available). One complexity that arises in this case is that repetitions of a stimulus are processed by the brain somewhat differently than the first presentation of the stimulus.
>
> ### **Q6: fitting of zone residuals**
>
> Please see response to **Q4**.
>
> ### **Q7: simulations of misspecified settings**
>
> **Summary:** We agree with the reviewer that adding additional simulations to discuss plausible alternate models such as when $Y_{1}$ is a cause of $Y_{2}$ will strengthen the paper. We have implemented two alternate data generation models where $Y_{1}$ is a cause of $Y_{2}$ and evaluated if our proposed framework can help us infer if a stimulus affects two brain zones in the same way in these scenarios. We find that when used together zone generalization and zone residuals enable us to infer if a stimulus affects two brain zones with significant encoding model performance in the same way.
>
> **Further details:** In the first model $X$ (the stimulus-representation) and $Z$ (the remaining stimulus information not captured by $X$) directly affect $Y_{2}$ and indirectly affect $Y_{2}$ through mediator $Y_{1}$:
> $$
> Y_{1, P} =
>     {g_{12, P}(X)} + \delta \times
>     h_{12, P}(Z) + (1-\delta) \times \epsilon_{12, P}
> $$
> $$
> Y_{2, P} = \tau \times Y_{1, P} +
>     {g_{2, P}(X)} + \delta \times
>     h_{2, P}(Z) + (1-\delta) \times \epsilon_{2, P}
> $$
>
> We introduce an adjustable parameter $\tau$ to simulate a wide range of scenarios. Parameter $\tau \in [-1, 1]$ controls the extent and direction that $Y_{2}$’s measurements are affected by $Y_{1}$. The remaining parameters and functions were previously defined in Appendix B.1.
>
> In the second model $X$ and $Z$ only indirectly affect $Y_{2}$ through mediator $Y_{1}$:
> $$ Y_{1, P} =
>     {g_{12, P}(X)} + \delta \times h_{12, P}(Z) + (1-\delta) \times \epsilon_{12, P} $$
> $$ Y_{2, P} = \tau \times Y_{1, P} +
>     \epsilon_{2, P}
> $$
>
> We evaluated the three metrics in our framework on synthetically generated data from both models similarly to Fig. 2. Open Review does not support adding images directly in the response, so we have included the new figures via an anonymized link: https://drive.google.com/file/d/142m0_efSIIn-YcP9BQFRvs3fq15g7qcV/view?usp=sharing
> For both data generation models we find that only zone generalization is informative of how similar two brain zones respond to stimulus properties, enabling us to determine which pair of inferences (A or B vs. C or D) can be made (Rebuttal Figs. 4-5 left). We also find that for the first model only zone residuals vary when the proportion of stimulus properties that are driving the brain zones but not captured by the stimulus-representation changes (Rebuttal Figs. 4 right). For the second model, we find that when we vary the proportion of stimulus properties that are driving the brain zones but not captured by the stimulus-representation the zone residuals remain constant at $0$ (Rebuttal Fig. 5 right). This is expected as the stimulus properties that drive the brain zones are shared between the two zones. Therefore, for the second model only inference B or C is possible. This suggests that for both models, we can use zone generalizations to narrow our search down to a pair of inferences (A or B vs. C or D), and then zone residuals can be used to precisely infer how the stimulus affects the pair of zones.
>
> ### **Q8: reasons for asymmetric metrics**
>
> The asymmetry in both zone generalization and zone residuals can be due to a difference in SNR, and also to a difference in the proportions of all stimulus properties that similarly affect the zones. To illustrate this, consider the extreme case in which the stimulus properties that affect zone_1 are a strict subset of those that affect zone_2 and the effect on the two zones is similar. Then, zone generalization(zone1, zone2) < zone generalization(zone2, zone1) and zone residuals(zone1, zone2) < zone residuals(zone2, zone1). Disentangling this from the SNR effect on asymmetry is an interesting direction for future work. We will clarify this in the text.

---

### Official Review · Reviewer_wCe1 · 2021-11-22

**Confidence:** 3
**Overall Score:** 6

**Main Review:**

Pros:
1. the considered problem is interesting and important.
2. the real data application in the paper is interesting.
3. Although without any theoretical/mathematical justification, the proposed metrics are intuitively reasonable.

Cons:
1. Section 2 of this paper is a bit of confusing to me, because three different models are presented in this section, and the main metrics, zone generalisation and zone residuals, are defined based on different models. So it is unclear to me about which is the model that you main method (Figure 1) based on?
2. Although the considered problem has some "causal flavour", I feel that the proposed method is rather irrelevant to causality. So I am not sure whether this paper would be a good fit for this conference.

**Summary:**

This paper considers the problem of whether the stimulus properties affect different brain zones in the same way. To address this problem, the authors proposed a method (Figure 1 in the paper) which involves two new metrics: zone generalisation and zone residuals. Synthetic simulations and real data applications are implemented to illustrate the performance of the proposed method.

---

> ### Author Response · Authors · 2021-12-04
> **Response to reviewer wCe1**
>
> We thank the reviewer for their feedback and believe that incorporating the clarifications in response to the reviewer’s comments will strengthen the manuscript. We respond to the reviewer's comments below.
>
> ### **Q1: model fitting**
>
> **Summary:** We fit all three metrics presented in Fig 1 based on the most general and most commonly-assumed model (encoding models and zone generalization are fit based on Eq1; zone residuals are fit based on Eq1 when the stimulus representation X is replaced by the brain recordings for a second brain zone Y_j).
>
> **Further details:** However, the true underlying model of how brain activity is generated as a function of the stimulus is of course unknown. In the remainder of Sec 2, we posit that there can be additional components in the brain activity, which are important for understanding whether two zones are affected by stimuli in a similar or different way. We introduce the two additional models (Eq2 & Eq4) to conceptually break down the independent component epsilon in Eq1 into these additional important components. Our simulations show that when these additional components are present in the brain data, our proposed metrics (zone generalization and zone residuals): 1) are indeed sensitive to these additional components, despite being fit based on the most general model in Eq 1, and 2) allow us to make one of the four main inferences in Fig 1, when used together. We thank the reviewer for pointing out this confusing part and we will clarify it in the paper.
>
> ### **Q2: venue fit**
>
> We ask the reviewer to see our response to reviewer *Nc86* (specifically to **Q4**), which more formally establishes the framework of our paper and hopefully reflects its causal aspects better.
>
> Further, we are trying to build connections between the causal world and computational cognitive neuroscience. Our goal is to more formally establish the framework of encoding model analysis so we can begin to address issues such as what do encoding models truly capture, how much can we say about how a stimulus affects the activity in a brain region using an encoding model, what to do with inter-feature correlation, how to attribute specific information to different features when they are correlated, etc. All these concerns have always been part of the encoding model approach in neuroscience, and neuroscientists have sometimes tried to address them, but have often either simply acknowledged them or completely evaded them. We hope that featuring this paper at CLeaR will spark further connections between computational cognitive neuroscience and the world of causality. Discussing the paper in the CLeaR conference will benefit neuroscientists by bringing in ideas and feedback from causality researchers. It will also be useful for causality researchers to see what questions are open at the moment in computational cognitive neuroscience, and to understand the gap between theoretical methods for testing causal inference and the real, practical setting where they can be used, so that one day this gap can be bridged.

---

### Official Review · Reviewer_Nc86 · 2021-11-22

**Confidence:** 5
**Overall Score:** 7

**Main Review:**

This is a well written paper proposing a quantitative framework to expand the inferences that can be done in the study of the causal effects of task stimuli on brain signals. The originality of the proposal is based on the explicit modeling of a unique and shared task causal effect (which the zone generalization metric aims to capture), and in the explicit modeling of unique and shared real task features causal effects not captured by the chosen task stimuli model (ie. difference between stimulus properties and stimulus representation) (which the zone residuals metric aims to capture). The manuscript contains a supplementary material file with useful code to track simulations and analysis. The supplementary material also contains a useful guideline explaining how the metrics can be normalized according to the research problem at hand and a very useful figure (Fig. 6) illustrating the relationship between brain zones task-evoked effects, stimulus properties and stimulus representations. Some points require clarification:

If you apply a standard task linear regression for zones Y1 and Y2: Y1 = bT + e1, where b, that measures the causal effect of T on Y1, is statistically significant, and Y2 = aT + e2, where a is also statistically significant and  a != b, it seems that I could claim that the same task representation T produced different causal effects on zones Y1 and Y2, given that a!=b. Similarly if a == b, I could claim that task representation T produced the same causal effect on zone Y1 and Y2. How is this inference different from what you measure with the zone generalization metric?

The zone residuals metric takes into account the number of possible pairs of participants. Does the metric approximate to its real value when the number of participants in the dataset tends to infinity? Or, can the metric have a bias problem if we do not have enough participants?

A fundamental point of the paper is the ability to make the four inferences proposed by the framework based on assessments of large and small values of the metrics. In the empirical application the authors used a cutoff based on simulations, and in the discussion this point is pushed to future research. I consider that given the importance of this cutoff to make the correct inference, more should be discussed in the manuscript about how the selection of the cutoff could be done, for example data-driven, previous-knowledge, comparison to other tasks. This of course will be only ideas, but they could give the readers a sense of how this problem can be solved. In my view, without a way to solve this problem the framework loses utility.

The two metrics, zone generalization and zone residuals, are pairwise measures. Can they be thought from a multivariate perspective? For example, by controlling for the effect of a third zone? In other words, can a third zone, for example, confound the metrics, as it may occur in functional connectivity analysis?


**Summary:**

The authors present two metrics to make inference about how two different brain zones are affected by the same experimental stimulus. The first metric, zone generalization, is the correlation between the predicted task-evoked signal for one zone and the actual measured task-evoked signal for another zone. The second metric, zone residuals, is the average across pairs of participants of the correlation between the residuals of regressing the actual task-evoked signal of one zone from another zone, for each participant in a pair. Using these metrics the authors propose four types of inferences that combine conclusions about the similarity of the effect of one stimulus on two different brain zones and about stimulus-related effects not captured by the stimulus model used. The authors illustrate the application of the metrics on simulations that manipulate the degree of similarity of response and the degree of stimulus properties not captured by the stimulus model. (In the supplementary material an analysis of signal to noise ratio is also presented.) The authors then apply the metrics to two empirical fMRI movie watching datasets and 34 language regions of interest, illustrating the four different inferences allowed by their approach. In the supplementary material the authors show in simulations that functional connectivity and some implementations of representational similarity analysis do not allow for the inferences proposed in this framework.

---

> ### Author Response · Authors · 2021-12-04
> **Response to reviewer Nc86, Part 2 of 2**
>
> Continued from Part 1
>
> ### **Q4: addressing possible confounders from additional brain zones**
>
> **Summary:** In the case of functional connectivity analysis where the researcher is interested in causal discovery (i.e., estimating a causal DAG for the brain regions), we agree that it is important to account for other zones that might confound the discovered relationship between two zones. However, our framework focuses on inferring if the causal effect of the stimulus on two zones is the same. In response to the reviewer's question, we showed with do-calculus that the introduction of a third zone does not change how we can estimate the causal effect of the stimulus on each brain zone. We provide more details about this analysis below.
>
> **Further details:** In our framework, we are interested in $P(Y_{1} = y_{1} |do(S= s))$ and $P(Y_{2} = y_{2} |do(S=s))$, where $Y_{i}$ represents the observation at brain zone $i$ and $S$ is the stimulus. In the case where there are only edges between both $S$ and $Y_{1}$ and $S$ and $Y_{2}$, these terms of interest would be equivalent to $P(Y_{1} = y_{1} |S = s)$ and $P(Y_{2} = y_{2} |S = s)$. We show in this link https://drive.google.com/file/d/1dCt3vffgywel8VVJ40y_Ek_BB6KK3DiC/view?usp=sharing (and in the response to reviewer czAT which includes simulations) that this is also the case if there is an additional edge between $Y_{1}$ and $Y_{2}$. To test if the introduction of a third zone confounds our metrics we consider possible configurations for causal directed acyclic graphs (DAGs) with four nodes, 3 brain zone nodes and 1 stimulus node under our experimental paradigm and assumptions (previously specified in the Introduction). Using do-calculus we find that in each case the probability that a brain zone ($Y_{i}$) equals a specific value ($Y_{i} = y_{i}$) conditioned on intervening on the stimulus ($S$) is $P(Y_{i} = y_{i} | do(S=s)) = P(Y_{i} = y_{i} | S=s)$ (proofs are available in link:https://drive.google.com/file/d/1dCt3vffgywel8VVJ40y_Ek_BB6KK3DiC/view?usp=sharing). This demonstrates that, under our assumptions (also specified in the link https://drive.google.com/file/d/1dCt3vffgywel8VVJ40y_Ek_BB6KK3DiC/view?usp=sharing and paper), the introduction of a third zone does not change the way we can estimate the causal effect of the stimulus on each brain zone. (We can still use an encoding model to estimate $P(Y_{i}|S)$, we do not need more complex models that account for other zones.) Therefore, we do not think that the presence of other zones confounds our metrics. We will add a discussion of this point to the paper. Now that we made the argument above, we have to add that in order to eventually fully understand brain function, it will be necessary (in future work) to model and understand the relationship between all the different brain zones that are affected by a given stimulus.

---

> > ### Author Response · Authors · 2021-12-04
> > **Response to reviewer Nc86, Part 1 of 2**
> >
> > We thank the reviewer for their thoughtful comments and positive initial evaluation. We believe that incorporating their feedback will strengthen the manuscript. We attempt to address the reviewer's questions below.
> >
> > ### **Q1: zone generalization vs $a==b$**
> >
> > When T is high-dimensional and has possibly correlated features, it becomes difficult to interpret the difference between a and b. This is the case in our setting so we instead find it useful to measure the distance between a and b in the final transformed space ($\hat{a}T$ vs $\hat{b}T$). Zone generalization is in fact proportional to this distance (i.e. $corr(\hat{Y_1}, \hat{Y_2})$), in the limit of sufficient data (i.e. when the encoding model is close to the Bayes-optimal predictor). Further, zone generalization looks not only at how much the weights are different, but at how much this difference affects the ability to predict held-out data. It could be considered a more stringent method, ignoring small differences in weights that don’t affect generalization performance. We would be happy to include a discussion of this relationship in the text if the reviewer would find it useful.
> >
> > ### **Q2: zone residuals and number of participants**
> >
> > **Summary:** In response to the reviewer’s question, we evaluate the stability of zone residuals with increasing numbers of participants. The zone residuals’ stability increases with increasing sample size until it stabilizes with a sample size of 5-10 participants. Therefore, we recommend using zone residuals for datasets with at least five participants. We will include this recommendation in the text.
> >
> > **Further details:** We evaluated the stability of the zone residuals metric by calculating this metric for 88 sample sizes (ranging from 2 to 89 participants) from the HCP dataset. For each sample size $n$ we randomly sampled $n$ participants 100 times, each time sampling without replacement. Then we calculated the zone residuals between each pair of brain zones. As it is difficult to visualize the results for all ROI pairs we present the average zone residual value for each sample size for the ROI pairs shown in Fig. 6. Open Review does not support adding images directly in the response, so we have included the new figures via an anonymized link: https://drive.google.com/file/d/18oXni0MvKIgNc2AGfLC-fHjxTIEobgJT/view?usp=sharing. These results demonstrate that the zone residual metric approximates it’s real value when the number of participants in the dataset tends to infinity when there are at least five participants.
> >
> > ### **Q3: selecting cutoffs**
> >
> > We agree with the reviewer that selecting the cutoff is an important part of applying the framework to real data. One idea is to make the decisions based on significance, similarly to the first step of the framework. Because there aren’t good priors for the chance values of zone generalization and zone residuals, the null distributions can be estimated using permutation tests. There are several parts of this analysis that are nontrivial and may influence the final outcome (e.g., permutation tests at the group-level or a participant-level, and how to aggregate results across participants), and these can be investigated in depth in future works. We will include this suggestion in the text.
> >
> > Independently from these future efforts, we do believe that our current work makes important contributions by showing that common tools for inferring relationships between stimuli and brain activity are limited and to characterize these limitations, especially when considering complex stimulus representations (i.e. representations with multiple sources of variance). While more work is needed to achieve the perfect inference framework, we do believe our contributions reveal an important issue and present a significant step towards resolving it, and that these contributions are timely, considering the recent interest in relating complex stimuli representations obtained from deep neural networks to brain recordings (Jain and Huth, 2018; Toneva and Wehbe, 2019; Wang et al., 2019; Schrimpf et al., 2020; Caucheteux et al., 2021b; Goldstein et al., 2021; Cross et al., 2021).

---

### Decision · Program_Chairs · 2022-01-12

**Decision:**

Accept (Poster)

**Comment:**

The authors develop two metrics for evaluating how different brain zones may be affected by the same stimuli. The framework created shows promising results when applied to actual and simulated data. This is a well-written paper with a particular application in mind. It designs a framework for discussing causal effects for applications in neuroscience. This paper's developed metrics and framework are potentially impactful and present a promising start for future research in this direction. The reviewers and I found the author's replies and rebuttal especially helpful in reaching our decision, so I hope some of these comments and arguments are included in the final version of the paper.

Minor comment: In addition to the comments made by the reviewers below, I would encourage the authors to make the references more consistent. At the moment some authors' first names are included and others are not.